# Improved Sample Complexity for Reward-free Reinforcement Learning under Low-rank MDPs

**Yuan Cheng**[*]
University of Science and Technology of China
cy16@mail.ustc.edu.cn

**Ruiquan Huang**[*]
The Pennsylvania State University
rzh5514@psu.edu

**Jing Yang**
The Pennsylvania State University
yangjing@psu.edu

**Yingbin Liang**
The Ohio State University
liang.889@osu.edu

## Abstract

In reward-free reinforcement learning (RL), an agent explores the environment first without any reward information, in order to achieve certain learning goals afterwards for any given reward. In this paper we focus on reward-free RL under low-rank MDP models, in which both the representation and linear weight vectors are unknown. Although various algorithms have been proposed for reward-free low-rank MDPs, the corresponding sample complexity is still far from being satisfactory. In this work, we first provide the first known sample complexity lower bound that holds for any algorithm under low-rank MDPs. This lower bound implies it is strictly harder to find a near-optimal policy under low-rank MDPs than under linear MDPs. We then propose a novel model-based algorithm, coined RAFFLE, and show it can both find an $\epsilon$-optimal policy and achieve an $\epsilon$-accurate system identification via reward-free exploration, with a sample complexity significantly improving the previous results. Such a sample complexity matches our lower bound in the dependence on $\epsilon$, as well as on $K$ in the large $d$ regime, where $d$ and $K$ respectively denote the representation dimension and action space cardinality. Finally, we provide a planning algorithm (without further interaction with true environment) for RAFFLE to learn a near-accurate representation, which is the first known representation learning guarantee under the same setting.

## 1 Introduction

Reward-free reinforcement learning, recently formalized by Jin et al. (2020b), arises as a powerful framework to accommodate diverse demands in sequential learning applications. Under the reward-free RL framework, an agent first explores the environment without reward information during the exploration phase, with the objective to achieve certain learning goals later on for any given reward function during the planning phase. Such a learning goal can be to find an $\epsilon$-optimal policy, to achieve an $\epsilon$-accurate system identification, etc. The reward-free RL paradigm may find broad application in many real-world engineering problems. For instance, reward-free exploration can be efficient when various reward functions are taken into consideration over a single environment, such as safe RL (Miryoosefi & Jin, 2021; Huang et al., 2022), multi-objective RL (Wu et al., 2021), multi-task RL (Agarwal et al., 2022; Cheng et al., 2022), etc. Studies of reward-free RL on the theoretical side have been largely focused on characterizing the sample complexity to achieve a learning goal under various MDP models. Specifically, reward-free tabular RL has been studied in Jin et al. (2020a); Ménard et al. (2021); Kaufmann et al. (2021); Zhang et al. (2020). For reward-free RL with function approximation, Wang et al. (2020) studied *linear MDPs* introduced by Jin et al. (2020b), where both the transition and the reward are linear functions of a given feature extractor, Zhang et al.

---

[*]Equal contribution

(2021b) studied *linear mixture MDPs* introduced by Ayoub et al. (2020), and Zanette et al. (2020b) considered a classes of MDPs with *low inherent Bellman error* introduced by Zanette et al. (2020a).

In this paper, we focus on reward-free RL under low-rank MDPs, where the transition kernel admits a decomposition into two embedding functions that map to low dimensional spaces. Compared with linear MDPs, the feature functions (i.e., the representation) under low-rank MDPs are unknown, hence the design further requires representation learning and becomes more challenging. Reward-free RL under low-rank MDPs was first studied by Agarwal et al. (2020), and the authors introduced a provably efficient algorithm FLAMBE, which achieves the learning goal of system identification with a sample complexity of $\tilde{O}(\frac{H^{22}K^9 d^7}{\epsilon^{10}})$. Here $d$, $H$ and $K$ respectively denote the representation dimension, episode horizon, and action space cardinality. Later on, Modi et al. (2021) proposed a model-free algorithm MOFFLE for reward-free RL under low-nonnegative-rank MDPs (where feature functions are non-negative), for which the sample complexity for finding an $\epsilon$-optimal policy scales as $\tilde{O}(\frac{H^5 K^5 d^3_{LV}}{\epsilon^2 \eta})$ (which is rescaled under the condition of $\sum_{h=1}^{H} r_h \leq 1$ for fair comparison). Here, $d_{LV}$ denotes the non-negative rank of the transition kernel, which may be exponentially larger than $d$ as shown in Agarwal et al. (2020), and $\eta$ denotes the positive reachability probability to all states, where $1/\eta$ can be as large as $\sqrt{d_{LV}}$ as shown in Uehara et al. (2022b). Recently, a reward-free algorithm called RFOLIVE has been proposed under non-linear MDPs with low Bellman Eluder dimension (Chen et al., 2022b), which can be specialized to low-rank MDPs. However, RFOLIVE is computationally more costly and considers a special reward function class, making their complexity result not directly comparable to other studies on reward-free low-rank MDPs.

This paper investigates reward-free RL under low-rank MDPs to address the following important open questions:

- For low-rank MDPs, none of previous studies establishes a lower bound on the sample complexity showing a necessary sample complexity requirement for near-optimal policy finding.
- The sample complexity of previous algorithms in Agarwal et al. (2020); Modi et al. (2021) on reward-free low-rank MDP is polynomial in the involved parameters, but still much higher than desirable. It is vital to improve the algorithm to further reduce the sample complexity.
- Previous studies on low-rank MDPs did not provide estimation accuracy guarantee on the learned representation (only on the transition kernels). However, such a representation learning guarantee can be very beneficial to reuse the learned representation in other RL environment.

## 1.1 MAIN CONTRIBUTIONS

We summarize our main contributions in this work below.

- **Lower bound:** We provide the first-known lower bound $\tilde{\Omega}(\frac{HdK}{\epsilon^2})$ on the sample complexity that holds for any algorithm under the same low-rank MDP setting. Our proof lies in a novel construction of hard MDP instances that capture the necessity of the cardinality of the action space on the sample complexity. Interestingly, comparing this lower bound for low-rank MDPs with the upper bound for linear MDPs in Wang et al. (2020) further implies that it is strictly more challenging to find near-optimal policy under low-rank MDPs than linear MDPs.
- **Algorithm:** We propose a new model-based reward-free RL algorithm under low-rank MDPs. The central idea of RAFFLE lies in the construction of a novel exploration-driven reward, whose corresponding value function serves as an upper bound on the model estimation error. Hence, such a pseudo-reward encourages the exploration to collect samples over those state-action space where the model estimation error is large so that later stage of the algorithm can further reduce such an error based on those samples. Such reward construction is new for low-rank MDPs, and serve as the key reason for our improved sample complexity.
- **Sample complexity:** We show that our algorithm can both find an $\epsilon$-optimal policy and achieve an $\epsilon$-accurate system identification via reward-free exploration, with a sample complexity of $\tilde{O}(\frac{H^3 d^2 K(d^2+K)}{\epsilon^2})$, which matches our lower bound in terms of the dependence on $\epsilon$ as well as on $K$ in the large $d$ regime. Our result significantly improves that of $\tilde{O}(\frac{H^{22}K^9 d^7}{\epsilon^{10}})$ in Agarwal et al. (2020) to achieve the same goal. Our result also improves the sample complexity of $\tilde{O}(\frac{H^5 K^5 d^3_{LV}}{\epsilon^2 \eta})$ in Modi et al. (2021) in three aspects: order on $K$ is reduced; $d$ can be exponentially smaller than $d_{LV}$ as shown in Agarwal et al. (2020); and no introduction of $\eta$, where $1/\eta$

can be as large as $\sqrt{d_{LV}}$. Further, our result on reward-free RL naturally achieves the goal of reward-known RL, which improves that of $\tilde{O}\left(\frac{H^5 d^4 K^2}{\epsilon^2}\right)$ in Uehara et al. (2022b) by $\Theta(H^2)$.

- **Near-accurate representation learning:** We design a planning algorithm that exploits the exploration phase of RAFFLE to further learn a provably near-accurate representation of the transition kernel without requiring further interaction with the environment. To the best of our knowledge, this is the first theoretical guarantee on representation learning for low-rank MDPs.

## 2 PRELIMINARIES AND PROBLEM FORMULATION

**Notation.** For any $H \in \mathbb{N}$, we denote $[H] := \{1, \ldots, H\}$. For any vector $x$ and symmetric matrix $A$, we denote $\|x\|_2$ as the $\ell_2$ norm of $x$ and $\|x\|_A := \sqrt{x^\top A x}$. For any matrix $A$, we denote $\|A\|_F$ as its Frobenius norm and let $\sigma_i(A)$ be its $i$-th largest singular value. For two probability measures $P, Q \in \Omega$, we use $\|P - Q\|_{TV}$ to denote their total variation distance.

### 2.1 EPISODIC MDPS

We consider an episodic Markov decision process (MDP) $\mathcal{M} = (\mathcal{S}, \mathcal{A}, H, P, r)$, where $\mathcal{S}$ can be an arbitrarily large state space; $\mathcal{A}$ is a finite action space with cardinality $K$; $H$ is the number of steps in each episode; $P : \mathcal{S} \times \mathcal{A} \times \mathcal{S} \to [0, 1]$ is the time-dependent transition kernel, where $P_h(s_{h+1}|s_h, a_h)$ denotes the transition probability from the state-action pair $(s_h, a_h)$ at step $h$ to state $s_{h+1}$ in the next step; $r_h : \mathcal{S} \times \mathcal{A} \to [0, 1]$ denotes the deterministic reward function at step $h$; We further normalize the summation of reward function as $\sum_h^H r_h \leq 1$. A policy $\pi$ is a set of mappings $\{\pi_h : \mathcal{S} \to \Delta(\mathcal{A})\}_{h \in [H]}$, where $\Delta(\mathcal{A})$ is the set of all probability distributions over the action space $\mathcal{A}$. Further, $a \sim \mathcal{U}(\mathcal{A})$ indicates the uniform selection of an action $a$ from $\mathcal{A}$.

In each episode of the MDP, we assume that a fixed initial state $s_1$ is drawn. Then, at each step $h \in [H]$. the agent observes state $s_h \in \mathcal{S}$, takes an action $a_h \in \mathcal{A}$ under a policy $\pi_h$, and receives a reward $r_h(s_h, a_h)$ (in the reward-known setting), and then the system transits to the next state $s_{h+1}$ with probability $P_h(s_{h+1}|s_h, a_h)$. The episode ends after $H$ steps.

As standard in the literature, we use $s_h \sim (P, \pi)$ to denote a state sampled by executing the policy $\pi$ under the transition kernel $P$ for $h - 1$ steps. If the previous state-action pair $(s_{h-1}, a_{h-1})$ is given, we use $s_h \sim P$ to denote that $s_h$ follows the distribution $P_h(\cdot|s_{h-1}, a_{h-1})$. We use the notation $\mathbb{E}_{(s_h, a_h) \sim (P, \pi)}[\cdot]$ to denote the expectation over states $s_h \sim (P, \pi)$ and actions $a_h \sim \pi$.

For a given policy $\pi$ and an MDP $\mathcal{M} = (\mathcal{S}, \mathcal{A}, H, P, r)$, we denote the value function starting from state $s_h$ at step $h$ as $V_{h,P,r}^\pi(s_h) := \mathbb{E}_{(s_{h'}, a_{h'}) \sim (P, \pi)}\left[\sum_{h'=h}^H r_{h'}(s_{h'}, a_{h'})|s_h\right]$. We use $V_{P,r}^\pi$ to denote $V_{1,P,r}^\pi(s_1)$ for simplicity. Similarly, we denote the action-value function starting from state action pair $(s_h, a_h)$ at step $h$ as $Q_{h,P,r}^\pi(s_h, a_h) := r_h(s_h, a_h) + \mathbb{E}_{s_{h+1} \sim P}\left[V_{h+1,P,r}^\pi(s_{h+1})|s_h, a_h\right]$.

We use $P^\star$ to denote the transition kernel of the true environment and for simplicity, denote $\mathbb{E}_{(s_h, a_h) \sim (P^\star, \pi)}[\cdot]$ as $\mathbb{E}_\pi^\star[\cdot]$. Given a reward function $r$, there always exists an optimal policy $\pi^\star$ that yields the optimal value $V_{P^\star, r}^\star = \sup_\pi V_{P^\star, r}^\pi$.

### 2.2 LOW-RANK MDPS

This paper focuses on the low-rank MDPs (Agarwal et al., 2020) defined as follows.

**Definition 1.** (Low-rank MDPs). A transition probability $P_h^\star : \mathcal{S} \times \mathcal{A} \to \Delta(\mathcal{A})$ admits a low-rank decomposition with dimension $d \in \mathbb{N}$ if there exists two embedding functions $\phi_h^\star : \mathcal{S} \times \mathcal{A} \to \mathbb{R}^d$ and $\mu_h^\star : \mathcal{S} \to \mathbb{R}^d$ such that $P_h^\star(s'|s, a) = \langle \phi_h^\star(s, a), \mu_h^\star(s') \rangle, \forall s, s' \in \mathcal{S}, a \in \mathcal{A}$.

For normalization, we assume $\|\phi_h^\star(s, a)\|_2 \leq 1$ for all $(s, a)$, and for any function $g : \mathcal{S} \to [0, 1], \|\int \mu_h^\star(s)g(s)ds\|_2 \leq \sqrt{d}$. An MDP $\mathcal{M}$ is a low-rank MDP with dimension $d$ if for each $h \in [H]$, $P_h$ admits a low-rank decomposition with dimension $d$. We use $\phi^\star = \{\phi_h^\star\}_{h \in [H]}$ and $\mu^\star = \{\mu_h^\star\}_{h \in [H]}$ to denote the embeddings for $P^\star$.

We remark that when $\phi_h$ is revealed to the agent, low-rank MDPs specialize to linear MDPs (Wang et al., 2020; Jin et al., 2020b). Essentially, low-rank MDPs do not assume that the features $\{\phi_h\}_h$

are known a priori. The lack of knowledge on features in fact invokes a nonlinear structure, which makes the model strictly harder than linear MDPs or tabular models. Since it is impossible to learn a model in polynomial time if there is no assumption on features $\phi_h$ and $\mu_h$, we adopt the following conventional assumption from the recent studies on low-rank MDPs.

**Assumption 1.** (Realizability). A learning agent can access to a model class $\{(\Phi, \Psi)\}$ that contains the true model, i.e., the embeddings $\phi^\star \in \Phi, \mu^\star \in \Psi$.

While we assume the cardinality of the function classes to be finite for simplicity, extensions to infinite classes with bounded statistical complexity (such as bounded covering number) are not difficult (Sun et al., 2019; Agarwal et al., 2020).

### 2.3 REWARD-FREE RL AND LEARNING OBJECTIVES

Reward-free RL typically has two phases: exploration and planning. In the *exploration* phase, an agent explores the state space via interaction with the true environment and can collect samples over multiple episodes, but without access to the reward information. In the *planning* phase, the agent is no longer allowed to interact with the environment, and for any given reward function, is required to achieve certain learning goals (elaborated below) based on the outcome of the exploration phase.

The planning phase may require the agent to achieve different learning goals. In this paper, we focus on three of such goals. The most popular goal in reward-free RL is to find a near-optimal policy that achieves the best value function under the true environment with $\epsilon$-accuracy, as defined below.

**Definition 2.** ($\epsilon$-optimal policy). Fix $\epsilon > 0$. For any given reward function $r$, a learned policy $\pi$ is $\epsilon$-optimal if it satisfies $V_{P^\star,r}^{\pi^\star} - V_{P^\star,r}^{\pi} \le \epsilon$.

For model-based learning, Agarwal et al. (2020) proposed *system identification* as another useful learning goal, defined as follows.

**Definition 3** ($\epsilon$-accurate system identification). Fix $\epsilon > 0$. Given a model class $(\Phi, \Psi)$, a learned model $(\hat{\phi}, \hat{\mu})$ is said to achieve $\epsilon$-accurate system identification if it uniformly approximates the true model $P^\star$, i.e., $\forall \pi, h \in [H], \mathbb{E}_\pi^\star \left[ \left\| \left\langle \hat{\phi}_h(s_h, a_h), \hat{\mu}_h(\cdot) \right\rangle - P_h^\star(\cdot|s_h, a_h) \right\|_{TV} \right] \le \epsilon$.

Besides those two common learning goals, we also propose an additional goal on near-accurate representation learning. Towards that, we introduce the following divergence-based metric to quantify distance between two representations, which has been used in supervised learning (Du et al., 2021b).

**Definition 4.** (Divergence between two representations). Given a distribution $q$ over $\mathcal{S} \times \mathcal{A}$ and two representations $\phi, \phi' \in \Phi$, define the covariance between $\phi$ and $\phi'$ w.r.t $q$ as $\Sigma_{(s,a)\sim q}(\phi, \phi') = \mathbb{E}\left[\phi(s,a)\phi'(s,a)^\top\right]$. Then, the divergence between $\phi$ and $\phi'$ with respect to $q$ is defined as

$$D_q(\phi, \phi') = \Sigma_{(s,a)\sim q}(\phi', \phi') - \Sigma_{(s,a)\sim q}(\phi', \phi) \left(\Sigma_{(s,a)\sim q}(\phi, \phi)\right)^\dagger \Sigma_{(s,a)\sim q}(\phi, \phi').$$

It can be verified that $D_q(\phi, \phi') \succeq 0$ (i.e., positive semidefinite) and $D_q(\phi, \phi) = 0$ for any $\phi, \phi' \in \Phi$.

## 3 LOWER BOUND ON SAMPLE COMPLEXITY

In this section, we provide a lower bound on the sample complexity that all reward-free RL algorithms must satisfy under low-rank MDPs. The detailed proof can be found in Appendix C.

**Theorem 1** (Lower bound). *For any algorithm that can output an $\epsilon$-optimal policy (as Definition 2), if $H > \max(24\epsilon, 4), S \ge 6, K \ge 3$ and $\delta < 1/16$, then there exists a low-rank MDP model $\mathcal{M}$ such that the number of trajectories sampled by the algorithm is at least $\Omega\left(\frac{HdK}{\epsilon^2}\right)$.*

To the best of our knowledge, Theorem 1 establishes the first lower bound for learning low-rank MDPs in the reward-free setting. More importantly, Theorem 1 shows that it is strictly more costly in terms of sample complexity to find near-optimal policies under *low-rank* MDPs (which have unknown representations) than *linear* MDPs (which have known representations) by at least a factor of $\Omega(K)$. This can be seen as the lower bound in Theorem 1 for low-rank MDPs has an additional term $K$ compared to the upper bound $\tilde{O}\left(\frac{d^3 H^4}{\epsilon^2}\right)$ provided in Wang et al. (2020) for linear MDPs.

This can be explained intuitively as follows. In linear models, all the representations $\phi : \mathcal{S} \times \mathcal{A} \to \mathbb{R}^d$ are known. Then, it requires at most $O(d)$ actions with linearly independent features to realize all transitions $\langle \phi, \mu \rangle$. However, learning low-rank MDPs requires the agent to further select $O(K)$ actions to access the unknown features, leading to a dependence on $K$.

Our proof of the new lower bound mainly features the following two novel ingredients in the construction of hard MDP instances. a) We divide the actions into two types. The first type of actions is mainly used to form a large state space through a tree structure. The second type of actions is mainly used to distinguish different MDPs. Such a construction allows us to separately treat the state space and the action space, so that both state space and action space can be arbitrarily large. b) We explicitly define the feature vectors for all state-action pairs, and more importantly, the dimension is less than or equal to the number of states. These two ingredients together guarantee that the number of actions $K$ can be arbitrarily large and independent with other parameters $d$ and $S$, which indicates that the dependence on the number of actions $K$ is unavoidable.

## 4 THE RAFFLE ALGORITHM

In this section, we propose RAFFLE (see Algorithm 1) for reward-free RL under low-rank MDPs.

**Summary of design novelty:** The central idea of RAFFLE lies in the construction of a novel exploration-driven reward, which is desirable because its corresponding value function serves as an upper bound on the model estimation error during the exploration phase. Hence, such a pseudo-reward encourages the exploration to collect samples over those state-action space where the model estimation error is large so that later stage of the algorithm can further reduce such an error based on those samples. Such reward construction are new for low-rank MDPs, and serve as the key enabler for our improved sample complexity. They also necessitate various new ingredients in other steps of algorithms, as elaborated below.

**Exploration and MLE model estimation.** In each iteration $n$ during the exploration phase, for each $h \in [H]$, the agent executes the exploration policy $\pi_{n-1}$ (defined in the previous iteration) up to step $h-1$, after which it takes two uniformly selected actions, and stops after step $h+1$. Different from FLAMBE (Agarwal et al., 2020) that collects a large number of samples for each episode, our algorithm uses each exploration policy to collect only one sample trajectory at each episode, indexed by $(n, h)$. Hence, the sample complexity of RAFFLE is much smaller than that of FLAMBE. In fact, such an efficient sampling together with our new termination idea introduced later benefit sample complexity.

Then, the agent estimates the low-rank components $\hat{\phi}_h^{(n)}$ and $\hat{\mu}_h^{(n)}$ via the MLE oracle with given model class $(\Phi, \Psi)$ and a dataset $\mathcal{D}_h^n$ as follows

$$\text{MLE}(\mathcal{D}_h^n) := \arg\max_{\phi \in \Phi, \mu \in \Psi} \sum_{(s,a,s') \in \mathcal{D}_h^n} \log \langle \phi_h(s,a), \mu_h(s') \rangle.$$

**Design of exploration reward.** The agent updates the empirical covariance matrix $\hat{U}_h^{(n)}$ as

$$\hat{U}_h^{(n)} = \lambda_n I + \sum_{\tau=1}^n \hat{\phi}_h^{(n)}(s_h^{(\tau,h+1)}, a_h^{(\tau,h+1)})(\hat{\phi}_h^{(n)}(s_h^{(\tau,h+1)}, a_h^{(\tau,h+1)}))^\top, \tag{1}$$

where $\{s_h^{(\tau,h+1)}, a_h^{(\tau,h+1)}, s_{h+1}^{(\tau,h+1)}\}$ is collected at iteration $\tau$, episode $(h+1)$, and step $h$.

Next, the agent uses both $\hat{\phi}_h^{(n)}$ and $\hat{U}_h^{(n)}$ to construct an *exploration-driven reward* function as

$$\hat{b}_h^{(n)}(s,a) = \min\{\hat{\alpha}_n \|\hat{\phi}_h^{(n)}(s,a)\|_{(\hat{U}_h^{(n)})^{-1}}, 1\}, \tag{2}$$

where $\hat{\alpha}_n$ is a pre-determined parameter. We note that although individual $\hat{b}_h^{(n)}(s,a)$ for each step may not represent point-wise uncertainty as indicated in Uehara et al. (2022b), we find its total cumulative version $\hat{V}_{\hat{P}^{(n)}, \hat{b}^{(n)}}^{\pi}$ can serve as a *trajectory-wise* uncertainty measure to select exploration policy. To see this, it can be shown that for any $\pi$ and $h$, $\mathbb{E}_\pi^\star \left[ \left\| \left\langle \hat{\phi}_h^{(n)}(s_h, a_h), \hat{\mu}_h^{(n)}(\cdot) \right\rangle - P_h^\star(\cdot | s_h, a_h) \right\|_{TV} \right] \leq c' \hat{V}_{\hat{P}^{(n)}, \hat{b}^{(n)}}^{\pi_n} + \sqrt{\frac{c_n}{n}}$, where $c'$ is a constant and $c_n = O(\log n)$. As iteration number $n$ grows, the second term diminishes to zero, which indicates that $\hat{V}_{\hat{P}^{(n)}, \hat{b}^{(n)}}^{\pi_n}$ (under the reward of $\hat{b}^{(n)}$) serves as a good upper bound on the estimation error

for the true transition kernel. Hence, exploration guided by maximizing $\hat{V}^{\pi}_{\hat{P}^{(n)}, \hat{b}^{(n)}}$ will collect more trajectories over which the learned transition kernels are not estimated well. This will help to reduce the model estimation error in the future.

---

**Algorithm 1 RAFFLE (R**ew**A**rd-**F**ree **F**eature **LE**arning**)**

1: **Input:** $\hat{\alpha}_n, \zeta_n, \epsilon > 0, \delta \in (0, 1)$, regularizer $\lambda_n$, model classes $\{(\mu, \phi) : \mu \in \Psi, \phi \in \Phi\}$.
2: Initialize $\pi_0(\cdot|s)$ to be uniform; set $\mathcal{D}_h^0 = \emptyset$.
3: **Phase I: Exploration Phase**
4: **for** $n = 1, \ldots$ **do**
5:     **for** $h = 1, \ldots, H$ **do**
6:         Use $\pi_{n-1}$: roll into $s_{h-1}$, uniformly choose $a_{h-1}, a_h$, enter into $s_h, s_{h+1}$.
7:         Collect data $s_1^{(n,h)}, a_1^{(n,h)}, \ldots, s_h^{(n,h)}, a_h^{(n,h)}, s_{h+1}^{(n,h)}$.
8:         Add the triple $(s_h^{(n,h)}, a_h^{(n,h)}, s_{h+1}^{(n,h)})$ to the dataset $\mathcal{D}_h^n = \mathcal{D}_h^{n-1} \cup \{(s_h^{(n,h)}, a_h^{(n,h)}, s_{h+1}^{(n,h)})\}$.
9:         Learn $(\hat{\phi}_h^{(n)}, \hat{\mu}_h^{(n)}) = \text{MLE}(\mathcal{D}_h^n)$.
10:        Update transition dynamics $\hat{P}^{(n)}$ as $\hat{P}_h^{(n)}(s'|s, a) = \langle \hat{\phi}_h^{(n)}(s, a), \hat{\mu}_h^{(n)}(s') \rangle$.
11:     **end for**
12:     Update empirical covariance matrix $\hat{U}_h^{(n)}$ as in Equation (1).
13:     Define exploration-driven reward function $\hat{b}_h^{(n)}$ as in Equation (2).
14:     Define an estimated value function $\hat{V}^{\pi}_{\hat{P}^{(n)}, \hat{b}^{(n)}}$ based on $\hat{P}^{(n)}$ and $\hat{b}^{(n)}$ as in Equation (3).
15:     Find exploration policy $\pi_n = \arg\max_\pi \hat{V}^{\pi}_{\hat{P}^{(n)}, \hat{b}^{(n)}}$.
16:     **if** $2\hat{V}^{\pi_n}_{\hat{P}^{(n)}, \hat{b}^{(n)}} + 2\sqrt{K\zeta_n} \leq \epsilon$ **then**
17:         Terminate **Phase I: Exploration Phase** and set $\hat{P}^{\epsilon} = \hat{P}^{(n)}, \hat{b}^{\epsilon} = \hat{b}^{(n)}, \pi_{\epsilon} = \pi_n, n_{\epsilon} = n$.
18:     **end if**
19: **end for**
20: **Phase II: Planning Phase**
21: **Option 1** (learn near-optimal policy): Receive reward function $r = \{r_h\}_{h=1}^H$, and compute policy $\bar{\pi} = \arg\max_\pi V^{\pi}_{\hat{P}^{\epsilon}, r}$.
22: **Option 2** (system identification): let $\hat{P} = \hat{P}^{\epsilon}$.
23: **Option 3** (learn near-accurate representation): call Algorithm 2 of RepLearn and obtain $\tilde{\phi}$.
24: **Output:** policy $\bar{\pi}$, learned transition dynamics $\hat{P}^{\epsilon}$, learned representation $\tilde{\phi}$.

---

**Design of exploration policy.** The agent defines a *truncated value function* iteratively using the estimated transition kernel and the exploration-driven reward as follows:

$$\hat{Q}^{\pi}_{h, \hat{P}^{(n)}, \hat{b}^{(n)}}(s_h, a_h) = \min \left\{ 1, \hat{b}_h^{(n)}(s_h, a_h) + \hat{P}_h^{(n)} \hat{V}^{\pi}_{h+1, \hat{P}^{(n)}, \hat{b}^{(n)}}(s_h, a_h) \right\},$$

$$\hat{V}^{\pi}_{h, \hat{P}^{(n)}, \hat{b}^{(n)}}(s_h) = \mathbb{E}_{\pi} \left[ \hat{Q}^{\pi}_{h, \hat{P}^{(n)}, \hat{b}^{(n)}}(s_h, a_h) \right]. \tag{3}$$

The truncation technique here is important for the improvement of the sample complexity on the dependence of $H$. The agent finally finds an optimal policy maximizing $\hat{V}^{\pi}_{\hat{P}^{(n)}, \hat{b}^{(n)}}$, and uses this policy as the exploration policy for the next iteration.

**Novel termination criterion.** RAFFLE does not require a pre-determined maximum number of iterations as its input. Instead, it will **terminate** and output the current estimated model if the optimal value function $\hat{V}^{\pi}_{\hat{P}^{(n)}, \hat{b}^{(n)}}$ plus a minor term is below a threshold. Such a termination criterion essentially guarantees that the value functions under the estimated and true models are close to each other under any reward and policy, hence the exploration can be terminated in finite steps. Such a termination criterion enables our algorithm to identify an accurate model and later find a near-optimal policy with fewer sample collections than FLAMBE in Agarwal et al. (2020) as we discuss in the exploration phase. Additionally, our termination criterion provides strong performance guarantees on the output policy and estimator from the last iteration. On the contrary, Uehara et al. (2022b) can only provide guarantees on a random mixture of the policies obtained over all iterations.

**Planning phase.** Given any reward function $r$, the agent finds a near-optimal policy by planning with the learned transition dynamics $\hat{P}^\epsilon$ and the given reward $r$. Note that such planning with a known low-rank MDP is computationally efficient by assumption.

## 5 UPPER BOUNDS ON SAMPLE COMPLEXITY

In this section, we first show that the policy returned by RAFFLE is an $\epsilon$-optimal policy with respect to any given reward $r$ in the planning phase. The detailed proof can be found in Appendix A.

**Theorem 2** ($\epsilon$-optimal policy). *Assume $\mathcal{M}$ is a low-rank MDP with dimension $d$, and Assumption 1 holds. Given any $\epsilon, \delta \in (0, 1)$, and any reward function $r$, let $\bar{\pi}$ and $\hat{P}^\epsilon$ be the output of RAFFLE and $\pi^\star := \arg\max_\pi V_{P^\star,r}^\pi$ be the optimal policy under the true model $P^\star$. Set $\hat{\alpha}_n = \tilde{O}(\sqrt{K + d^2})$ and $\lambda_n = \tilde{O}(d)$. Then, with probability at least $1 - \delta$, we have $V_{P^\star,r}^{\pi^\star} - V_{P^\star,r}^{\bar{\pi}} \leq \epsilon$, and the total number of trajectories collected by RAFFLE is upper bounded by $\tilde{O}(\frac{H^3 d^2 K (d^2 + K)}{\epsilon^2})$.*

We note that the upper bound in Theorem 2 matches the lower bound in Theorem 1 in terms of the dependence on $\epsilon$ as well as on $K$ in the large $d$ regime.

Compared with MOFFLE (Modi et al., 2021), which also finds an $\epsilon$-optimal policy in reward-free RL, our result improves their sample complexity of $\tilde{O}\left(\frac{H^5 K^5 d_{LV}^3}{\epsilon^2 \eta}\right)$ in three aspects. First, the order on $K$ is reduced. Second, the dimension $d_{LV}$ of the underlying function class in MOFFLE can be exponentially larger than $d$ as shown in Agarwal et al. (2020). Finally, MOFFLE requires reachability assumption, leading to a factor $1/\eta$ in the sample complexity, which can be as large as $\sqrt{d_{LV}}$. Further, Theorem 2 naturally achieves the goal of reward-known RL with the same sample complexity, which improves that of $\tilde{O}\left(\frac{H^5 d^4 K^2}{\epsilon^2}\right)$ in Uehara et al. (2022b) by a factor of $O(H^2)$.

Proceeding to the learning objective of system identification, the learned transition kernel output by Algorithm 1 achieves the goal of $\epsilon$-accurate system identification with the same sample complexity as follows. The detailed proof can be found in Appendix B.

**Theorem 3** ($\epsilon$-accurate system identification). *Under the same condition of Theorem 2 and let $\hat{P}^\epsilon = \{\hat{\phi}_h^\epsilon, \hat{\mu}_h^\epsilon\}$ be the output of RAFFLE. Then, with probability at least $1 - \delta$, $\hat{P}^\epsilon$ achieves $\epsilon$-accurate system identification, i.e. for any $\pi$ and $h$: $\mathbb{E}_\pi^\star \left[\left\| \left\langle \hat{\phi}_h^\epsilon(s_h, a_h), \hat{\mu}_h^\epsilon(\cdot) \right\rangle - P_h^\star(\cdot | s_h, a_h) \right\|_{TV}\right] \leq \epsilon$, and the number of trajectories collected by RAFFLE is upper bounded by $\tilde{O}(\frac{H^3 d^2 K (d^2 + K)}{\epsilon^2})$.*

Theorem 3 significantly improves the sample complexity of $\tilde{O}(\frac{H^{22} K^9 d^7}{\epsilon^{10}})$ in Agarwal et al. (2020) on the dependence of all involved parameters for achieving $\epsilon$-accurate system identification.

## 6 NEAR-ACCURATE REPRESENTATION LEARNING

In low-rank MDPs, it is of great interest to learn the representation $\phi$ accurately, because other similar RL environments can very likely share the same representation (Rusu et al., 2016; Zhu et al., 2020; Dayan, 1993) and hence such learned representation can be directly reused in those environments. Thus, the third objective of RAFFLE in the planning phase is to provide an accurate estimation of $\phi$. We note that although RAFFLE provides an estimation of $\phi$ during its execution, such an estimation does not come with an accuracy guarantee. Besides, Theorem 3 on system identification does not provide the guarantee on the representation $\hat{\phi}$, but only on the entire transition kernel $\hat{P}$. Further, none of previous studies of reward-free RL under low-rank MDPs (Agarwal et al., 2020; Modi et al., 2021; Uehara et al., 2022b) established the guarantee on $\hat{\phi}$.

### 6.1 THE REPLEARN ALGORITHM

In this section, we present the following algorithm of RepLearn, which exploits the learned transition kernel from RAFFLE and learns a near-accurate representation without additional interaction with the environment. The formal version of RepLearn, Algorithm 2, is delayed in Appendix D. We explain the main idea of Algorithm 2 as follows. First, for each $h \in [H], t \in [T]$, where $T$ is the

number of rewards, $N_f$ pairs of state-action $(s_h, a_h)$ are generated based on distribution $q_h$. Note that the agent does not interact with the true environment during such data generation. Then, for any $h$, if we set the reward $r$ at step $h$ to be zero, $Q^\pi_{P^\star, h, r}(s_h, a_h)$ can have a linear structure in terms of the true representation $\phi^\star_h(s_h, a_h)$. Namely, there exists a $w_h$ decided by $r$, $P^\star$ and $\pi$ such that $Q^\pi_{P^\star, h, r}(s_h, a_h) = \langle \phi^\star_h(s_h, a_h), w_h \rangle$. Then, with the estimated transition kernel $\hat{P}$ that RAFFLE provides and the well-designed rewards $r^{h,t}$, $Q^{\pi^t}_{\hat{P}, h, r^{h,t}}(s_h, a_h)$ can be computed efficiently and serve as a target to learn the representation via the following regression problem:

$$\arg\min_{\phi_h \in \Phi, w^t_h \in \mathbb{R}^d} \sum_{t \in [T]} \sum_{(s_h, a_h) \in \mathcal{D}^t_h} (Q^{\pi^t}_{\hat{P}, h, r^{h,t}}(s_h, a_h) - \langle \phi_h(s_h, a_h), w^t_h \rangle)^2. \quad (4)$$

The main difference between our algorithm and that in Lu et al. (2021) for representation learning is that, our data generation is based on $\hat{P}$ from RAFFLE, which carries a natural estimation error but requires no interaction with the environment, whereas their algorithm assumes a generative model to collect data from the ground-truth transition kernel.

## 6.2 GUARANTEE ON ACCURACY

In order to guarantee the learned representation by Algorithm 2 is sufficiently close to the ground truth, we need to employ the following two somewhat necessary assumptions.

First, for the distributions of the state-action pairs $\{q_h\}^H_{h=1}$ in Algorithm 2, it is desirable to have $\hat{P}(\cdot|s, a)$ approximates true $P^\star(\cdot|s, a)$ well over those distributions, so that $Q^\pi_{\hat{P}, h, r}(s_h, a_h)$ can approximate the ground truth well. Intuitively, if some state-action pairs can hardly be visited under any policy, the output $\hat{P}(\cdot|s, a)$ of Algorithm 1 cannot approximate true $P^\star(\cdot|s, a)$ well over these state-action pairs. Hence, we assume reachability type assumptions for MDPs as following so that all state-action pairs is likely to be visited by certain policy. Such reachability type assumption is common in relevant literature (Modi et al., 2021; Agarwal et al., 2020).

As discussed in Section 6, intuitively, if some state action pairs can be hardly visited by any policy, the output of Algorithm 1 $\hat{P}(\cdot|s, a)$ can not approximate true $P^\star(\cdot|s, a)$ well over these state action pairs, so a standard reachability type assumption is necessary so that all states can be visited.

**Assumption 2** (Reachability). For the true transition kernel $P^\star$, there exists a policy $\pi^0$ such that $\min_{s \in \mathcal{S}} \mathbb{P}^{\pi^0}_h(s) \geq \eta_{\min}$, where $\mathbb{P}^{\pi^0}_h(\cdot) : \mathcal{S} \to \mathbb{R}$ is the density function over $\mathcal{S}$ using policy $\pi^0$ to roll into state $s$ at timestep $h$.

We further assume the input distributions $\{q_h\}^H_{h=1}$ are bounded with constant $C_B$. Then, together with Assumption 2, for any $(s, a) \in \mathcal{S} \times \mathcal{A}$, we have $q_h(s, a) \leq C_{\min} \mathbb{P}^{\pi^0}_h(s, a)$, where $C_{\min} = \frac{C_B}{\eta_{\min}}$.

Next, we assume that the rewards chosen for generating target $Q$-functions are sufficiently diverse so that the target $Q^\pi_{P^\star, h, r}(s_h, a_h)$ spans over the entire representation space to guarantee accurate representation learning in Equation (4). Such an assumption is commonly adopted in multi-task representation learning literature (Du et al., 2021b; Yang et al., 2021; Lu et al., 2021). To formally state the assumption, for any $h \in [H]$, let $\{r^{h,t}\}_{t \in [T]}$ be a set of $T$ rewards (where $T \geq d$) satisfying $r^{h,t}_h = 0$. As a result, for each $t$, given $\pi^t$, there exists $w^{t\,\star}_h \in \mathbb{R}^d$ such that $Q^\pi_{P^\star, h, r^{h,t}}(s_h, a_h) = \langle \phi^\star_h(s_h, a_h), w^{t\,\star}_h \rangle$. Let $W^\star_h = [w^{1\,\star}_h, \ldots, w^{T\,\star}_h] \in \mathbb{R}^{d \times T}$.

**Assumption 3** (Diverse rewards). The smallest singular value $\sigma_d(W^\star_h)$ of $W^\star_h$ defined above satisfies $\sigma^2_d(W^\star_h) \geq \Omega(\frac{T}{d})$, i.e., there exists a constant $C_D > 0$ such that $\sigma^2_d(W^\star_h) \geq \frac{C_D T}{d}$.

We next characterize the accuracy of the output representation of Algorithm 2 in terms of the divergence Definition 4 between the learned and the ground truth representations in the following theorem and defer the proof to Appendix D.

**Theorem 4** (Guarantee for representation learning). *Under Assumption 1 and Assumption 3, for any $\epsilon, \delta \in (0, 1)$, any $h \in [H]$, and sufficiently large $N_f$, let $\hat{P}$ be the output transition kernel of RAFFLE that satisfies Theorem 3. With probability at least $1 - \delta$, the output $\tilde{\phi}$ of RAFFLE satisfies*

$$\left\| D_{q_h}(\phi^\star_h, \tilde{\phi}_h)^{1/2} \right\|^2_F = O\left( \frac{\epsilon d C_{\min}}{C_D} + \frac{d}{C_D} \sqrt{\frac{\log \frac{2}{\delta}}{T N_f}} \right). \quad (5)$$

We explain the bound in Equation (5) as follows. The first term arises from the system identification error $\epsilon$ by using the output of RAFFLE, and can be made as small as possible by choosing appropriate $\epsilon$. The second term is related to the randomness caused by sampling state-action pairs from input distribution $\{q_h\}_{h\in[H]}$, and vanishes as $N_f$ becomes large. Note that $N_f$ is the number of simulated samples in Algorithm 2, which does not require interaction with the true environment. Hence, it can be made sufficiently large to guarantee a small error.

Theorem 4 shows RAFFLE can learn a near-accurate representation, based on which learned representations can be further used in other RL environments sharing common representations, similarly in spirit to how representation learning has been exploited in supervised learning (Du et al., 2021b).

## 7 RELATED WORK

**Reward-free RL.** While various studies (Oudeyer et al., 2007; Bellemare et al., 2016; Burda et al., 2018; Colas et al., 2018; Nair et al., 2018; Eysenbach et al., 2018; Co-Reyes et al., 2018; Hazan et al., 2019; Du et al., 2019; Pong et al., 2019; Misra et al., 2020) proposed exploration algorithms for good coverage on the state space without using explicit reward signals, theoretically speaking, the paradigm of reward-free RL was first formalized by Jin et al. (2020a), where they provided both upper and lower bounds on the sample complexity. For tabular case, several follow-up studies (Kaufmann et al., 2021; Ménard et al., 2021) further improved the sample complexity, and Zhang et al. (2020) established the minimax optimality guarantee. Reward-free RL was also studied with function approximation. Wang et al. (2020) studied *linear MDPs*, and Zhang et al. (2021b) studied *linear mixture MDPs*. Further, Zanette et al. (2020b) considered a class of MDPs with *low inherent Bellman error* introduced by Zanette et al. (2020a). Chen et al. (2022b) proposed a reward-free algorithm called RFOLIVE under non-linear MDPs with low Bellman Eluder dimension. In addition, Miryoosefi & Jin (2021) proposed a reward-free approach to solving constrained RL problems with any given reward-free RL oracle under both the tabular and linear MDPs.

**Reward-free RL under low-rank MDPs.** As discussed in Section 1, reward-free RL under low-rank MDPs have been studied recently (Agarwal et al., 2020; Modi et al., 2021), and our result significant improves the sample complexity therein. When finite latent state space is assumed, low-rank MDPs specialize to block MDPs, under which algorithms termed as PCID (Du et al., 2019) and HOMER (Misra et al., 2020) achieved sample complexities of $\tilde{O}(\frac{d^4 H^2 K^4}{\min(\eta^4 \gamma_n^2, \epsilon^2)})$ and $\tilde{O}(\frac{d^8 H^4 K^4}{\min(\eta^3, \epsilon^2)})$, respectively. Our result on the general low-rank MDPs can be utilized to further improve those results for block MDPs.

**Reward-known RL under low-rank MDPs.** For reward-known RL, Uehara et al. (2022b) proposed a computationally efficient algorithm REP-UCB under low-rank MDPs. Our design of the reward-free algorithm for low-rank MDPs is inspired by their algorithm with several new ingredients as discussed in Section 4, and improves their sample complexity on the dependence of $H$. Meanwhile, algorithms have been proposed for MDP models with low Bellman rank (Jiang et al., 2017), low witness rank (Sun et al., 2019), bilinear classes (Du et al., 2021a) and low Bellman eluder dimension (Jin et al., 2021), which can specialize to low-rank MDPs. However, those algorithms are computationally more costly as remarked in Uehara et al. (2022b), although their sample complexity may have sharper dependence on $d, K$ or $H$. Specializing to block MDPs, Zhang et al. (2022) proposed an algorithm called BRIEE, which empirically achieves the state-of-art sample complexity for block MDP models. Besides, Zhang et al. (2021a) proposed an algorithm coined ReLEX for a slightly different low-rank model, and obtained a problem dependent regret upper bound.

## 8 CONCLUSIONS

In this paper, we investigate the reward-free reinforcement learning, in which the underlying model admits a low-rank structure. Without further assumptions, we propose an algorithm called RAFFLE which significantly improves the state-of-the-art sample complexity for accurate model estimation and near-optimal policy identification. We further design an algorithm to exploit the model learned by RAFFLE for accurate representation learning without further interaction with the environment. Although $\epsilon$-accurate system identification can easily induce $H\epsilon$-optimal policy, the relationship between these two learning goals under general reward-free exploration setting remain under-explored, and is an interesting topic for future investigation.

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

## A    PROOF OF THEOREM 2

We first provide a proof outline to highlight our key ideas in the analysis of Theorem 2, and then provide the detailed proof. To simplify the notation, we denote the total variation distance $\left\| \hat{P}_h^{(n)}(s_h, a_h) - P_h^\star(s_h, a_h) \right\|_{TV}$ by $f_h^{(n)}(s_h, a_h)$.

*Proof Outline for Theorem 2.* **Step 1** provides an upper bound on the difference of value functions under estimated model $\hat{P}^{(n)}$ and the true model $P^\star$ for any given policy $\pi$ and reward $r$, as given in the following proposition (see proposition 4 in appendix A.2).

**Proposition 1** (Informal). *There exist constants $c_n = O(\log n)$ and $c' = O(1)$ such that for any policy $\pi$ and reward $r$, with high probability, we have*

$$\left| V_{P^\star, r}^\pi - V_{\hat{P}^{(n)}, r}^\pi \right| \le \hat{V}_{\hat{P}^{(n)}, \hat{b}^{(n)}}^{\pi_n} + \sqrt{\frac{c_n}{n}}.$$

This proposition is inspired by Uehara et al. (2022b), while it generalizes to non-stationary setting and arbitrary reward scenario from infinite stationary MDP and fixed reward. The main proof idea is to first notice that for any reward, we have

$$\left| V_{P^\star, r}^\pi - V_{\hat{P}^{(n)}, r}^\pi \right| \le \hat{V}_{\hat{P}^{(n)}, f^{(n)}}^\pi. \tag{6}$$

Then, we show that $\hat{V}_{\hat{P}^{(n)}, f^{(n)}}^\pi \le \hat{V}_{\hat{P}^{(n)}, \hat{b}^{(n)}}^\pi + \sqrt{\frac{c_n}{n}} \le \hat{V}_{\hat{P}^{(n)}, \hat{b}^{(n)}}^{\pi_n} + \sqrt{\frac{c_n}{n}}$ due to the optimality of the exploration policy $\pi_n$.

**Step 2** shows sublinearity of the summation of $\hat{V}_{\hat{P}^{(n)}, \hat{b}^{(n)}}^{\pi_n}$, which is an upper bound of value function difference under the true model with the estimated model (see proposition 5 in appendix A.3).

**Proposition 2** (Informal). *Under the same setting of Proposition 1, with high probability, the summation of value functions $V_{\hat{P}^{(n)}, \hat{b}^{(n)}}^{\pi_n}$ under exploration policies $\{\pi_n\}_{n \in [N]}$ with exploration-driven reward functions $\hat{b}^{(n)}$ is sublinear, as given by*

$$\sum_{n \in [N]} \hat{V}_{\hat{P}^{(n)}, \hat{b}^{(n)}}^{\pi_n} \le \tilde{O}\left( Hd\sqrt{K(K + d^2)N} \right).$$

**Step 3** combines Step 2 and Step 1. We are able to conclude that RAFFLE will terminate with polynomial sample complexity such that, the value function difference of $P^\star$ and the returned environment $\hat{P}^\epsilon$ is at most $\epsilon$.

**Proposition 3.** *With high probability, RAFFLE terminates after at most $\tilde{O}\left( H^2 d^2 K(d^2 + K)/\epsilon^2 \right)$ iterations, and the output model $\hat{P}^\epsilon$ satisfies*

$$\left| V_{P^\star, r}^\pi - V_{\hat{P}^{(\epsilon)}, r}^\pi \right| \le \hat{V}_{\hat{P}^{(\epsilon)}, \hat{b}^{(n_\epsilon)}}^{\pi_{n_\epsilon}} + \sqrt{\frac{c_{n_\epsilon}}{n_\epsilon}} \le \epsilon/2,$$

*where $n_\epsilon$ is the iteration number where RAFFLE terminates, i.e. $\hat{P}^\epsilon = \hat{P}^{(n_\epsilon)}$.*

Finally, with some algebraic operations, Proposition 3 concludes the proof of Theorem 2. □

### A.1    SUPPORTING LEMMAS

We first present the high probability event.

**Lemma 1.** *We define $\Pi_n$ to be a uniform mixture of previous $n - 1$ exploration policies:*

$$\Pi_n = \mathcal{U}(\pi_0, \pi_1, ..., \pi_{n-1}).$$

*Denote the total variation of $\hat{P}^{(n)}$ and $P^\star$, and the expected matrix of $\hat{U}_h^{(n)}$ as follows.*

$$f_h^{(n)}(s_h, a_h) = \left\| P_h^\star(\cdot|s_h, a_h) - \hat{P}_h^{(n)}(\cdot|s_h, a_h) \right\|_{TV}, \tag{7}$$

$$U_{h,\phi}^{(n)} = n \mathop{\mathbb{E}}_{\substack{s_h \sim (P^\star, \Pi_n) \\ a_h \sim \mathcal{U}(\mathcal{A})}} \left[ \phi(s_h, a_h)(\phi(s_h, a_h))^\top \right] + \lambda_n I, \tag{8}$$

$$W_{h,\phi}^{(n)} = n \mathop{\mathbb{E}}_{(s_h, a_h) \sim (P^\star, \Pi_n)} \left[ \phi(s_h, a_h)(\phi(s_h, a_h))^\top \right] + \lambda_n I. \tag{9}$$

*where $\lambda_n = \beta_3 d \log(2nH|\Phi|/\delta))$ and $\beta_3 = O(1)$ is a constant coefficient. Suppose Algorithm 1 runs N iterations and let events $\mathcal{E}_0$ and $\mathcal{E}_1$ be defined as follows.*

$$\mathcal{E}_0 = \left\{ \forall n \in [N], h \in [H], s_h \in \mathcal{S}, a_h \in \mathcal{A}, \mathop{\mathbb{E}}_{\substack{s_h \sim (P^\star, \Pi_n) \\ a_h \sim \mathcal{U}}} \left[ f_h^{(n)}(s_h, a_h)^2 \right] \leq \zeta_n \right\},$$

$$\mathcal{E}_1 = \Bigg\{ \forall n \in [N], h \in [H], s_h \in \mathcal{S}, a_h \in \mathcal{A},$$

$$\frac{1}{5} \left\| \hat{\phi}_{h-1}^{(n)}(s,a) \right\|_{(U_{h-1,\hat{\phi}}^{(n)})^{-1}} \leq \left\| \hat{\phi}_{h-1}^{(n)}(s,a) \right\|_{(\hat{U}_{h-1}^{(n)})^{-1}} \leq 3 \left\| \hat{\phi}_{h-1}^{(n)}(s,a) \right\|_{(U_{h-1,\hat{\phi}}^{(n)})^{-1}} \Bigg\}.$$

*where $\zeta_n = \log(2|\Phi||\Psi|nH/\delta)/n$. We further denote $\zeta = N\zeta_N = \log(2|\Phi||\Psi|NH/\delta)$ for simplicity. Denote $\mathcal{E} = \mathcal{E}_0 \cap \mathcal{E}_1$ as the intersection of the two events. Then, $\mathbb{P}[\mathcal{E}] \geq 1 - \delta$.*

*Proof.* By Corollary 2 in Appendix F, we have $\mathbb{P}[\mathcal{E}_0] \geq 1 - \delta/2$. Further, by Lemma 39 in Zanette et al. (2021) for the version of fixed $\phi$ and Lemma 11 in Uehara et al. (2022b), we have $\mathbb{P}[\mathcal{E}_1] \geq 1 - \delta/2$. Therefore, $\mathbb{P}[\mathcal{E}] \geq 1 - \delta$. $\qquad\square$

Based on Lemma 1, we can bound the exlporation-driven reward in RAFFLE as follows.

**Corollary 1.** *Given that the event $\mathcal{E}$ occurs, the following inequality holds for any $n \in [N], h \in [H], s_h \in \mathcal{S}, a_h \in \mathcal{A}$:*

$$\min \left\{ \frac{\hat{\alpha}_n}{5} \left\| \hat{\phi}_h^{(n)}(s_h, a_h) \right\|_{(U_{h,\hat{\phi}}^{(n)})^{-1}}, 1 \right\} \leq \hat{b}_h^{(n)}(s_h, a_h) \leq 3\hat{\alpha}_n \left\| \hat{\phi}_h^{(n)}(s_h, a_h) \right\|_{(U_{h,\hat{\phi}}^{(n)})^{-1}},$$

*where $\hat{\alpha}_n = 5\sqrt{2\beta_3 n\zeta_n(K + d^2)}$.*

*Proof.* Recall $\hat{b}_h^{(n)}(s_h, a_h) = \min \left\{ \hat{\alpha}_n \left\| \hat{\phi}_h^{(n)}(s,a) \right\|_{(\hat{U}_h^{(n)})^{-1}}, 1 \right\}$. Applying Lemma 1, we can immediately obtain the result. $\qquad\square$

The following lemma extends the Lemmas 12 and 13 under infinite discount MDPs in Uehara et al. (2022b) to episodic MDPs. We provide the proof for completeness.

**Lemma 2.** *Let $P_{h-1} = \langle \phi_{h-1}, \mu_{h-1} \rangle$ be a generic MDP model, and $\Pi$ be an arbitrary and possibly mixture policy. Define an expected Gram matrix as follows*

$$M_{h-1,\phi} = \lambda_n I + n \mathop{\mathbb{E}}_{\substack{s_{h-1} \sim (P^\star, \Pi) \\ a_{h-1} \sim \Pi}} \left[ \phi_{h-1}(s_{h-1}, a_{h-1}) (\phi_{h-1}(s_{h-1}, a_{h-1}))^\top \right].$$

*Further, let $f_{h-1}(s_{h-1}, a_{h-1})$ be the total variation between $P_{h-1}^\star$ and $P_{h-1}$ at time step $h-1$. Suppose $g \in \mathcal{S} \times \mathcal{A} \to \mathbb{R}$ is bounded by $B \in (0, \infty)$, i.e., $\|g\|_\infty \leq B$. Then, $\forall h \geq 2, \forall$ policy $\pi_h$,*

$$\mathop{\mathbb{E}}_{\substack{s_h \sim P_{h-1} \\ a_h \sim \pi_h}} [g(s_h, a_h)|s_{h-1}, a_{h-1}]$$

$$\leq \|\phi_{h-1}(s_{h-1}, a_{h-1})\|_{(M_{h-1,\phi})^{-1}} \times$$

$$\sqrt{nK \mathop{\mathbb{E}}_{\substack{s_h \sim (P^\star, \Pi) \\ a_h \sim \mathcal{U}}} [g^2(s_h, a_h)] + \lambda_n dB^2 + nB^2 \mathop{\mathbb{E}}_{\substack{s_{h-1} \sim (P^\star, \Pi) \\ a_{h-1} \sim \Pi}} [f_{h-1}(s_{h-1}, a_{h-1})^2]}.$$

*Proof.* We first derive the following bound:

$$\mathbb{E}_{\substack{s_h \sim P_{h-1} \\ a_h \sim \pi_h}} [g(s_h, a_h)|s_{h-1}, a_{h-1}]$$

$$= \int_{s_h} \sum_{a_h} g(s_h, a_h)\pi(a_h|s_h)\langle\phi_{h-1}(s_{h-1}, a_{h-1}), \mu_{h-1}(s_h)\rangle ds_h$$

$$\leq \|\phi_{h-1}(s_{h-1}, a_{h-1})\|_{(M_{h-1,\phi})^{-1}} \left\| \int \sum_{a_h} g(s_h, a_h)\pi(a_h|s_h)\mu_{h-1}(s_h)ds_h \right\|_{M_{h-1,\phi}},$$

where the inequality follows from Cauchy-Schwarz inequality. We further expand the second term in the RHS of the above inequality as follows.

$$\left\| \int \sum_{a_h} g(s_h, a_h)\pi(a_h|s_h)\mu_{h-1}(s_h)ds_h \right\|_{M_{h-1,\phi}}^2$$

$$\overset{(i)}{\leq} n \mathbb{E}_{\substack{s_{h-1} \sim (P^\star, \Pi) \\ a_{h-1} \sim \Pi}} \left[ \left( \int_{s_h} \sum_{a_h} g(s_h, a_h)\pi_h(a_h|s_h)\mu(s_h)^\top\phi(s_{h-1}, a_{h-1})ds_h \right)^2 \right] + \lambda_n dB^2$$

$$= n \mathbb{E}_{\substack{s_{h-1} \sim (P^\star, \Pi) \\ a_{h-1} \sim \Pi}} \left[ \left( \mathbb{E}_{\substack{s_h \sim P_{h-1} \\ a_h \sim \pi_h}} \left[ g(s_h, a_h)\Big|s_{h-1}, a_{h-1} \right] \right)^2 \right] + \lambda_n dB^2$$

$$\overset{(ii)}{\leq} 2n \mathbb{E}_{\substack{s_{h-1} \sim (P^\star, \Pi) \\ a_{h-1} \sim \Pi}} \left[ \mathbb{E}_{\substack{s_h \sim P^\star_{h-1} \\ a_h \sim \pi_h}} \left[ g(s_h, a_h)\Big|s_{h-1}, a_{h-1} \right]^2 \right] + \lambda_n dB^2$$
$$+ 2nB^2 \mathbb{E}_{\substack{s_{h-1} \sim (P^\star, \Pi) \\ a_{h-1} \sim \Pi}} [f_{h-1}(s_{h-1}, a_{h-1})]^2$$

$$\overset{(iii)}{\leq} 2n \mathbb{E}_{\substack{s_{h-1} \sim (P^\star, \Pi) \\ a_{h-1} \sim \Pi}} \left[ \mathbb{E}_{\substack{s_h \sim P^\star_{h-1} \\ a_h \sim \pi_h}} \left[ g(s_h, a_h)^2\Big|s_{h-1}, a_{h-1} \right] \right] + \lambda_n dB^2$$
$$+ 2nB^2 \mathbb{E}_{\substack{s_{h-1} \sim (P^\star, \Pi) \\ a_{h-1} \sim \Pi}} \left[ f_{h-1}(s_{h-1}, a_{h-1})^2 \right]$$

$$\overset{(iv)}{\leq} 2nK \mathbb{E}_{\substack{s_h \sim (P^\star, \Pi) \\ a_h \sim \mathcal{U}}} \left[ g(s_h, a_h)^2 \right] + \lambda_n dB^2 + 2nB^2 \mathbb{E}_{\substack{s_{h-1} \sim (P^\star, \Pi) \\ a_{h-1} \sim \Pi}} \left[ f_{h-1}(s_{h-1}, a_{h-1})^2 \right],$$

where $(i)$ follows from the assumption that $\|g\|_\infty \leq B$, $(ii)$ is due to that $f_{h-1}(s_{h-1}, a_{h-1})$ is the total variation between $P^\star_{h-1}$ and $P_{h-1}$ at time step $h-1$ and the fact that $(a+b)^2 \leq 2a^2 + 2b^2$, $(iii)$ follows from Jensen's inequality, and $(iv)$ is due to importance sampling. This finishes the proof. $\qquad\square$

Based on Lemma 2, we summarize three useful inequalities which bridges the total variation $f_h^{(n)}$ and the exploration-driven reward $\hat{b}_h^{(n)}$.

**Lemma 3.** *Define*

$$W_{h,\phi}^{(n)} = n \mathbb{E}_{\substack{s_h \sim (P^\star, \Pi_n) \\ a_h \sim \Pi_n}} \left[ \phi(s_h, a_h)(\phi(s_h, a_h))^\top \right] + \lambda_n I, \tag{10}$$

*where $\lambda_n = \beta_3 d \log(2nH|\Phi|/\delta)$. Given that the event $\mathcal{E}$ occurs, the following inequalities hold. For any $n$, when $h \geq 2$,*

$$\mathbb{E}_{\substack{s_h \sim \hat{P}_{h-1}^{(n)} \\ a_h \sim \pi}}\left[f_h^{(n)}(s_h, a_h)\Big| s_{h-1}, a_{h-1}\right] \leq \alpha_n \left\|\hat{\phi}_{h-1}^{(n)}(s_{h-1}, a_{h-1})\right\|_{(U_{h-1,\hat{\phi}}^{(n)})^{-1}}, \tag{11}$$

$$\mathbb{E}_{\substack{s_h \sim P_{h-1}^* \\ a_h \sim \pi}}\left[f_h^{(n)}(s_h, a_h)\Big| s_{h-1}, a_{h-1}\right] \leq \alpha_n \left\|\phi_{h-1}^*(s_{h-1}, a_{h-1})\right\|_{(U_{h-1,\phi^\star}^{(n)})^{-1}}, \tag{12}$$

$$\mathbb{E}_{\substack{s_h \sim P_{h-1}^* \\ a_h \sim \pi}}\left[\hat{b}_h^{(n)}(s_h, a_h)\Big| s_{h-1}, a_{h-1}\right] \leq \gamma_n \left\|\phi_{h-1}^*(s_{h-1}, a_{h-1})\right\|_{(W_{h-1,\phi^\star}^{(n)})^{-1}}, \tag{13}$$

*where*

$$\alpha_n = \sqrt{2\beta_3 n \zeta_n(K + d^2)}, \quad \gamma_n = \sqrt{45\beta_3 n \zeta_n K d(K + d^2)}.$$

*Specially, when $h = 1$,*

$$\mathbb{E}_{a_1 \sim \pi}\left[f_1^{(n)}(s_1, a_1)\right] \leq \sqrt{K\zeta_n}, \qquad \mathbb{E}_{a_1 \sim \pi}\left[\hat{b}(s_1, a_1)\right] \leq 15\alpha_n \sqrt{\frac{dK}{n}}. \tag{14}$$

*Proof.* We start by developing Equation (11) as follows. Given that the event $\mathcal{E}$ occurs, for $h \geq 2$ we have

$$\mathbb{E}_{\substack{s_h \sim \hat{P}_{h-1}^{(n)} \\ a_h \sim \pi}}\left[f_h^{(n)}(s_h, a_h)\Big| s_{h-1}, a_{h-1}\right]$$

$$\overset{(i)}{\leq} \left\|\hat{\phi}_{h-1}^{(n)}(s_{h-1}, a_{h-1})\right\|_{(U_{h-1,\hat{\phi}}^{(n)})^{-1}} \times$$

$$\sqrt{nK \mathbb{E}_{\substack{s_{h-1} \sim (P^\star, \Pi_n) \\ a_{h-1}, a_h \sim \mathcal{U} \\ s_h \sim P^\star_h(\cdot|s_{h-1}, a_{h-1})}}[f_h^{(n)}(s_h, a_h)^2] + \lambda_n d + n \mathbb{E}_{\substack{s_{h-1} \sim (P^\star, \Pi_n) \\ a_{h-1} \sim \mathcal{U}}}\left[f_{h-1}^{(n)}(s_{h-1}, a_{h-1})^2\right]}$$

$$\overset{(ii)}{\leq} \left\|\hat{\phi}_{h-1}^{(n)}(s_{h-1}, a_{h-1})\right\|_{(U_{h-1,\hat{\phi}}^{(n)})^{-1}} \times$$

$$\sqrt{nK \mathbb{E}_{\substack{s_{h-1} \sim (P^\star, \Pi_n) \\ a_{h-1}, a_h \sim \mathcal{U} \\ s_h \sim P^\star_h(\cdot|s_{h-1}, a_{h-1})}}[f_h^{(n)}(s_h, a_h)^2] + \lambda_n d + nK \mathbb{E}_{\substack{s_{h-2} \sim (P^\star, \Pi_n) \\ a_{h-2}, a_{h-1} \sim \mathcal{U} \\ s_{h-1} \sim P^\star_{h-1}(\cdot|s_{h-2}, a_{h-2})}}\left[f_{h-1}^{(n)}(s_{h-1}, a_{h-1})^2\right]}$$

$$\overset{(iii)}{\leq} \left\|\hat{\phi}_{h-1}^{(n)}(s_{h-1}, a_{h-1})\right\|_{(U_{h-1,\hat{\phi}}^{(n)})^{-1}} \sqrt{2n\zeta_n K + \beta_3 n \zeta_n d^2}$$

$$\leq \alpha_n \left\|\hat{\phi}_{h-1}^{(n)}(s_{h-1}, a_{h-1})\right\|_{(U_{h-1,\hat{\phi}}^{(n)})^{-1}},$$

where $(i)$ follows from Lemma 2 and the fact that $f_h^{(n)}(s_h, a_h) \leq 1$, $(ii)$ follows from importance sampling at time step $h - 2$, and $(iii)$ follows from Lemma 1.

Equation (12) follows from the arguments similar to the above.

To obtain Equation (13), we first apply Lemma 2 and obtain

$$\mathbb{E}_{\substack{s_h \sim P_{h-1}^\star \\ a_h \sim \pi_n}}\left[\hat{b}_h^{(n)}(s_h, a_h)\Big| s_{h-1}, a_{h-1}\right]$$

$$\leq \left\|\phi_{h-1}^\star(s_{h-1}, a_{h-1})\right\|_{(W_{h-1,\phi^\star}^{(n)})^{-1}} \sqrt{nK \mathbb{E}_{\substack{s_h \sim (P^\star, \Pi_n) \\ a_h \sim \mathcal{U}}}[\{\hat{b}_h^{(n)}(s_h, a_h)\}^2] + \lambda_n d},$$

where we use the fact that $\hat{b}_h^{(n)}(s_h, a_h) \leq 1$. We further bound the term $n \mathbb{E}_{\substack{s_h \sim (P^\star, \Pi_n) \\ a_h \sim \mathcal{U}}} [(\hat{b}_h^{(n)}(s_h, a_h))^2]$ as follows:

$$
\begin{aligned}
&n \mathbb{E}_{\substack{s_h \sim (P^\star, \Pi_n) \\ a_h \sim \mathcal{U}}} \left[ \left( \hat{b}_h^{(n)}(s_h, a_h) \right)^2 \right] \\
&\leq n \mathbb{E}_{\substack{s_h \sim (P^\star, \Pi_n) \\ a_h \sim \mathcal{U}}} \left[ \hat{\alpha}_n^2 \left\| \hat{\phi}_h^{(n)}(s_h, a_h) \right\|_{(\hat{U}_{h,\hat{\phi}}^{(n)})^{-1}}^2 \right] \\
&\overset{(i)}{\leq} n \mathbb{E}_{\substack{s_h \sim (P^\star, \Pi_n) \\ a_h \sim \mathcal{U}}} \left[ 9\hat{\alpha}_n^2 \left\| \hat{\phi}_h^{(n)}(s_h, a_h) \right\|_{(U_{h,\hat{\phi}}^{(n)})^{-1}}^2 \right] \\
&= 9\hat{\alpha}_n^2 \mathrm{tr} \left\{ n \mathbb{E}_{\substack{s_h \sim (P^\star, \Pi_n) \\ a_h \sim \mathcal{U}}} \left[ \hat{\phi}_h^{(n)}(s_h, a_h) \hat{\phi}_h^{(n)}(s_h, a_h)^\top \left( n \mathbb{E}_{\substack{s_h \sim (P^\star, \Pi_n) \\ a_h \sim \mathcal{U}}} \left[ \hat{\phi}_h(s_h, a_h) \hat{\phi}_h^{(n)}(s_h, a_h)^\top \right] + \lambda_n I \right)^{-1} \right] \right\} \\
&\leq 9\hat{\alpha}_n^2 \mathrm{tr}(I) = 9\hat{\alpha}_n^2 d,
\end{aligned}
$$

where $(i)$ follows from Lemma 1, and we use $\mathrm{tr}(A)$ to denote the trace of any matrix $A$.

Hence,

$$
\mathbb{E}_{\substack{s_h \sim P_{h-1}^\star \\ a_h \sim \pi}} \left[ \hat{b}_h^{(n)}(s_h, a_h) \middle| s_{h-1}, a_{h-1} \right] \leq \left\| \phi_{h-1}^*(s_{h-1}, a_{h-1}) \right\|_{(W_{h-1,\phi^\star}^{(n)})^{-1}} \sqrt{9K\hat{\alpha}_n^2 d + \lambda_n d}
$$

$$
\leq \gamma_n \left\| \phi_{h-1}^*(s_{h-1}, a_{h-1}) \right\|_{W_{h-1,\phi^\star}^{(n)})^{-1}},
$$

where the last inequality follows from that $\hat{\alpha}_n = 5\alpha_n$ and the definition of $\gamma_n$ In addition, for $h = 1$, we have

$$
\mathbb{E}_{a_1 \sim \pi_n} \left[ f_1^{(n)}(s_1, a_1) \right] \overset{(i)}{\leq} \sqrt{K \mathbb{E}_{a_1 \sim \mathcal{U}} \left[ f_1^{(n)}(s_1, a_1)^2 \right]} \leq \sqrt{K\zeta_n},
$$

and

$$
\begin{aligned}
\mathbb{E}_{a_1 \sim \pi_n} \left[ \hat{b}(s_1, a_1) \right] &\overset{(ii)}{\leq} \hat{\alpha}_n \sqrt{K \mathbb{E}_{a_1 \sim \mathcal{U}} \left[ \|\hat{\phi}_1(s_1, a_1)\|_{(\hat{U}_{1,\hat{\phi}}^{(n)})^{-1}}^2 \right]} \\
&\leq 3\hat{\alpha}_n \sqrt{K \mathbb{E}_{a_1 \sim \mathcal{U}} \left[ \|\hat{\phi}_1(s_1, a_1)\|_{(U_{1,\hat{\phi}}^{(n)})^{-1}}^2 \right]} \\
&\leq 3\sqrt{\frac{25K\alpha_n^2 d}{n}} = 15\alpha_n \sqrt{\frac{dK}{n}},
\end{aligned}
$$

$\square$

## A.2 Proof of Proposition 1

Equipped with Lemma 3, the following proposition provides an upper bound on the difference of value functions under the estimated model $\hat{P}^{(n)}$ and the true model $P^\star$ for any given policy $\pi$ and reward $r$.

**Proposition 4** (Restatement of Proposition 1). *For all $n \in [N]$, policy $\pi$ and reward $r$, given that the event $\mathcal{E}$ occurs, we have*

$$
\left| V_{P^\star, r}^\pi - V_{\hat{P}^{(n)}, r}^\pi \right| \leq \hat{V}_{\hat{P}^{(n)}, \hat{b}^{(n)}}^\pi + \sqrt{K\zeta_n}.
$$

*Proof.* **Step 1.** We first show that $\left| V_{P^\star, r}^\pi - V_{\hat{P}^{(n)}, r}^\pi \right| \leq \hat{V}_{\hat{P}^{(n)}, f^{(n)}}^\pi$.

Recall the definition of estimated value functions $\hat{V}_{h,\hat{P}^{(n)},r}(s_h)$ and $\hat{Q}_{h,\hat{P}^{(n)},r}(s_h,a_h)$:

$$\hat{Q}^{\pi}_{h,\hat{P}^{(n)},r}(s_h,a_h) = \min\left\{1, r_h(s_h,a_h) + \hat{P}^{(n)}_h \hat{V}^{\pi}_{h+1,\hat{P}^{(n)},r}(s_h,a_h)\right\},$$

$$\hat{V}^{\pi}_{h,\hat{P}^{(n)},r}(s_h) = \mathbb{E}_{\pi}\left[\hat{Q}^{\pi}_{h,\hat{P}^{(n)},r}(s_h,a_h)\right].$$

We develop the proof by induction. For the base case $h = H + 1$, we have $\left|V^{\pi}_{H+1,\hat{P}^{(n)},r}(s_{H+1}) - V^{\pi}_{H+1,P^{\star},r}(s_{H+1})\right| = 0 = \hat{V}^{\pi}_{H+1,\hat{P}^{(n)},f^{(n)}}(s_{H+1})$.

Assume that $\left|V^{\pi}_{h+1,\hat{P}^{(n)},r}(s_{h+1}) - V^{\pi}_{h+1,P^{\star},r}(s_{h+1})\right| \le \hat{V}^{\pi}_{h+1,\hat{P}^{(n)},f^{(n)}}(s_{h+1})$ holds for any $s_{h+1}$.

Then, from Bellman equation, we have,

$$\left|Q^{\pi}_{h,\hat{P}^{(n)},r}(s_h,a_h) - Q^{\pi}_{h,P^{\star},r}(s_h,a_h)\right|$$

$$= \left|\hat{P}^{(n)}_h V^{\pi}_{h,\hat{P}^{(n)},r}(s_h,a_h) - P^{\star}_h V^{\pi}_{h+1,P^{\star},r}(s_h,a_h)\right|$$

$$= \left|\hat{P}^{(n)}_h \left(V^{\pi}_{h+1,\hat{P}^{(n)},r} - V^{\pi}_{h+1,P^{\star},r}\right)(s_h,a_h) + \left(\hat{P}^{(n)}_h - P^{\star}_h\right) V^{\pi}_{h,P^{\star},r}(s_h,a_h)\right|$$

$$\overset{(i)}{\le} \min\left\{1, f^{(n)}_h(s_h,a_h) + \hat{P}^{(n)}_h \left|V^{\pi}_{h+1,\hat{P}^{(n)},r} - V^{\pi}_{h+1,P^{\star},r}\right|(s_h,a_h)\right\}$$

$$\overset{(ii)}{\le} \min\left\{1, f^{(n)}_h(s_h,a_h) + \hat{P}^{(n)}_h \hat{V}^{\pi}_{h+1,\hat{P}^{(n)},f^{(n)}}(s_h,a_h)\right\}$$

$$= \hat{Q}^{\pi}_{h,\hat{P}^{(n)},f^{(n)}}(s_h,a_h), \tag{15}$$

where $(i)$ follows from the (action) value function is at most 1, and $(ii)$ follows from the induction hypothesis.

Then, by the definition of $\hat{V}^{\pi}_{h,\hat{P}^{(n)},r}(s_h)$, we have

$$\left|V^{\pi}_{h,\hat{P}^{(n)},r}(s_h) - V^{\pi}_{h,P^{\star},r}(s_h)\right|$$

$$= \left|\mathbb{E}_{\pi}\left[Q^{\pi}_{h,\hat{P}^{(n)},r}(s_h,a_h)\right] - \mathbb{E}_{\pi}\left[Q^{\pi}_{h,P^{\star},r}(s_h,a_h)\right]\right|$$

$$\le \mathbb{E}_{\pi}\left[\left|Q^{\pi}_{h,\hat{P}^{(n)},r}(s_h,a_h) - Q^{\pi}_{h,P^{\star},r}(s_h,a_h)\right|\right]$$

$$\overset{(i)}{\le} \mathbb{E}_{\pi}\left[\hat{Q}^{\pi}_{h,\hat{P}^{(n)},f^{(n)}}(s_h,a_h)\right]$$

$$= \hat{V}^{\pi}_{h,\hat{P}^{(n)},f^{(n)}}(s_h),$$

where $(i)$ follows from Equation (15).

Therefore, by induction, we have

$$\left|V^{\pi}_{P^{\star},r} - V^{\pi}_{\hat{P}^{(n)},r}\right| \le \hat{V}^{\pi}_{\hat{P}^{(n)},f^{(n)}}.$$

**Step 2.** Then, we show that $\hat{V}^{\pi}_{\hat{P}^{(n)},f^{(n)}} \le \hat{V}^{\pi}_{\hat{P}^{(n)},\hat{b}^{(n)}} + \sqrt{K\zeta_n}$.

By Equation (11) and the fact that the total variation distance is upper bounded by 1, with probability at least $1 - \delta/2$, we have

$$\mathbb{E}_{\hat{P}^{(n)},\pi}\left[f^{(n)}_h(s_h,a_h)\Big|s_{h-1}\right] \le \mathbb{E}_{a_{h-1}\sim\pi}\left[\min\left(\alpha_n\left\|\hat{\phi}^{(n)}_{h-1}(s_{h-1},a_{h-1})\right\|_{(U^{(n)}_{h-1,\hat{\phi}})^{-1}},1\right)\right], \forall h \ge 2. \tag{16}$$

Similarly, when $h = 1$,

$$\mathbb{E}_{a_1 \sim \pi} \left[ f_1^{(n)}(s_1, a_1) \right] \leq \sqrt{K \mathbb{E}_{a \sim \mathcal{U}} \left[ \left( f_1^{(n)}(s_1, a_1) \right)^2 \right]} \leq \sqrt{K \zeta_n}. \tag{17}$$

Based on Corollary 1, Equation (16) and $\alpha_n = 5\hat{\alpha}_n$, we have

$$\mathbb{E}_{\pi} \left[ \hat{b}_h^{(n)}(s_h, a_h) \Big| s_h \right] \geq \mathbb{E}_{\pi} \left[ \min \left( \alpha_n \left\| \hat{\phi}_h^{(n)}(s_h, a_h) \right\|_{(U_{h,\hat{\phi}}^{(n)})^{-1}}, 1 \right) \right] \geq \mathbb{E}_{\hat{P}^{(n)}, \pi} \left[ f_{h+1}^{(n)}(s_{h+1}, a_{h+1}) \Big| s_h \right]. \tag{18}$$

For the base case $h = H$, we have

$$\begin{aligned}
\mathbb{E}_{\hat{P}^{(n)}, \pi} \left[ \hat{V}_{H, \hat{P}^{(n)}, f^{(n)}}^{\pi}(s_H) \Big| s_{H-1} \right] &= \mathbb{E}_{\hat{P}^{(n)}, \pi} \left[ f_H^{(n)}(s_H, a_H) \Big| s_{H-1} \right] \\
&\leq \mathbb{E}_{\pi} \left[ b_{H-1}^{(n)}(s_{H-1}, a_{H-1}) | s_{H-1} \right] \\
&\leq \min \left\{ 1, \mathbb{E}_{\pi} \left[ \hat{Q}_{H-1, \hat{P}^{(n)}, \hat{b}^{(n)}}^{\pi}(s_{H-1}, a_{H-1}) \Big| s_{H-1} \right] \right\} \\
&= \hat{V}_{H-1, \hat{P}^{(n)}, \hat{b}^{(n)}}^{\pi}(s_{H-1}).
\end{aligned}$$

Assume that $\mathbb{E}_{\hat{P}^{(n)}, \pi} \left[ \hat{V}_{h+1, \hat{P}^{(n)}, f^{(n)}}^{\pi}(s_{h+1}) \Big| s_h \right] \leq \hat{V}_{h, \hat{P}^{(n)}, \hat{b}^{(n)}}^{\pi}(s_h)$ holds for step $h + 1$. Then, by Jensen's inequality, we obtain

$$\begin{aligned}
&\mathbb{E}_{\hat{P}^{(n)}, \pi} \left[ \hat{V}_{h, \hat{P}^{(n)}, f^{(n)}}^{\pi}(s_h) \Big| s_{h-1} \right] \\
&\leq \min \left\{ 1, \mathbb{E}_{\hat{P}^{(n)}, \pi} \left[ f_h^{(n)}(s_h, a_h) + \hat{P}_h^{(n)} \hat{V}_{h+1, \hat{P}^{(n)}, f^{(n)}}^{\pi}(s_h, a_h) \Big| s_{h-1} \right] \right\} \\
&\overset{(i)}{\leq} \min \left\{ 1, \mathbb{E}_{\pi} \left[ \hat{b}_{h-1}^{(n)}(s_{h-1}, a_{h-1}) \right] + \mathbb{E}_{\hat{P}^{(n)}, \pi} \left[ \mathbb{E}_{\hat{P}^{(n)}, \pi} \left[ \hat{V}_{h+1, \hat{P}^{(n)}, f^{(n)}}^{\pi}(s_{h+1}) \Big| s_h \right] \Big| s_{h-1} \right] \right\} \\
&\overset{(ii)}{\leq} \min \left\{ 1, \mathbb{E}_{\pi} \left[ b_{h-1}^{(n)}(s_{h-1}, a_{h-1}) \right] + \mathbb{E}_{\hat{P}^{(n)}, \pi} \left[ \hat{V}_{h, \hat{P}^{(n)}, \hat{b}^{(n)}}^{\pi}(s_h) \Big| s_{h-1} \right] \right\} \\
&= \min \left\{ 1, \mathbb{E}_{\pi} \left[ \hat{Q}_{h-1, \hat{P}^{(n)}, \hat{b}^{(n)}}^{\pi}(s_{h-1}, a_{h-1}) \right] \right\} \\
&= \hat{V}_{h-1, \hat{P}^{(n)}, \hat{b}^{(n)}}^{\pi}(s_{h-1}),
\end{aligned}$$

where $(i)$ follows from Equation (18), and $(ii)$ is due to the induction hypothesis.

By induction, we conclude that

$$\begin{aligned}
\hat{V}_{\hat{P}^{(n)}, f^{(n)}}^{\pi} &= \mathbb{E}_{\pi} \left[ f_1^{(s)}(s_1, a_1) \right] + \mathbb{E}_{\hat{P}^{(n)}, \pi} \left[ \hat{V}_{2, \hat{P}^{(n)}, f^{(n)}}^{\pi}(s_2) \Big| s_1 \right] \\
&\leq \sqrt{K \zeta_n} + \hat{V}_{\hat{P}^{(n)}, \hat{b}^{(n)}}^{\pi}.
\end{aligned}$$

Combining Step 1 and Step 2, we conclude that

$$\left| V_{P^\star, r}^{\pi} - V_{\hat{P}^{(n)}, r}^{\pi} \right| \leq \sqrt{K \zeta_n} + \hat{V}_{\hat{P}^{(n)}, \hat{b}^{(n)}}^{\pi}.$$

$\square$

### A.3   PROOF OF PROPOSITION 2

The following lemma is key to ensure that RAFFLE terminates in finite episodes.

**Proposition 5** (Restatement of Proposition 2). *Given that the event $\mathcal{E}$ occurs, $\zeta = \log\left(2|\Phi||\Psi|NH/\delta\right)$ the summation of the truncated value functions $\hat{V}^{\pi_n}_{\hat{P}^{(n)},\hat{b}^{(n)}}$ under exploration policies $\{\pi_n\}_{n\in[N]}$ is sublinear, i.e., the following bound holds:*

$$\sum_{n\in[N]} \hat{V}^{\pi_n}_{\hat{P}^{(n)},\hat{b}^{(n)}} + \sqrt{K\zeta_n} \leq 32\zeta Hd\sqrt{\beta_3 K(d^2+K)N}.$$

*Proof.* Note that $\hat{V}^{\pi}_{h,\hat{P}^{(n)},\hat{b}^{(n)}} \leq 1$ holds for any policy $\pi$ and $h\in[H]$. We first have

$$
\begin{aligned}
\hat{V}^{\pi_n}_{\hat{P}^{(n)},\hat{b}^{(n)}} - V^{\pi_n}_{P^\star,\hat{b}^{(n)}} &\leq \mathbb{E}_{\pi_n}\left[\hat{P}^{(n)}_1 \hat{V}^{\pi_n}_{2,\hat{P}^{(n)},\hat{b}^{(n)}}(s_1,a_1) - P^\star_1 V^{\pi_n}_{2,P^\star,\hat{b}^{(n)}}(s_1,a_1)\right] \\
&= \mathbb{E}_{\pi_n}\left[\left(\hat{P}^{(n)}_1 - P^\star_1\right)\hat{V}^{\pi_n}_{2,\hat{P}^{(n)},\hat{b}^{(n)}}(s_1,a_1) + P^\star_1\left(\hat{V}^{\pi_n}_{2,\hat{P}^{(n)},\hat{b}^{(n)}} - V^{\pi_n}_{2,P^\star,\hat{b}^{(n)}}\right)(s_1,a_1)\right] \\
&\leq \mathbb{E}_{\pi_n}\left[f^{(n)}_1(s_1,a_1) + P^\star_1\left(\hat{V}^{\pi_n}_{2,\hat{P}^{(n)},\hat{b}^{(n)}} - V^{\pi_n}_{2,P^\star,\hat{b}^{(n)}}\right)\right] \\
&\leq \cdots \\
&\leq \mathbb{E}_{(s_h,a_h)\sim(P^\star,\pi_n)}\left[\sum_{h=1}^H f^{(n)}(s_h,a_h)\right] = V^{\pi_n}_{P^\star,f^{(n)}},
\end{aligned}
$$

which implies $\hat{V}^{\pi_n}_{\hat{P}^{(n)},\hat{b}^{(n)}} \leq V^{\pi_n}_{P^\star,\hat{b}^{(n)}} + V^{\pi_n}_{P^\star,f^{(n)}}$.

Applying the Equation (13) and Equation (14), we obtain the following bound on the value function $V^{\pi_n}_{P^\star,\hat{b}^{(n)}}$:

$$
\begin{aligned}
V^{\pi_n}_{P^\star,\hat{b}^{(n)}} &= \sum_{h=1}^H \mathbb{E}_{\substack{s_h\sim(P^\star,\pi_n) \\ a_h\sim\pi_n}}\left[\hat{b}_n(s_h,a_h)\right] \\
&\leq \sum_{h=2}^H \mathbb{E}_{\substack{s_{h-1}\sim(P^\star,\pi_n) \\ a_{h-1}\sim\pi_n}}\left[\gamma_n\left\|\phi^\star_{h-1}(s_{h-1},a_{h-1})\right\|_{(W^{(n)}_{h-1,\phi^\star})^{-1}}\right] + 15\alpha_n\sqrt{\frac{dK}{n}} \\
&\leq \sum_{h=1}^H \mathbb{E}_{\substack{s_h\sim(P^\star,\pi_n) \\ a_h\sim\pi_n}}\left[\gamma_n\left\|\phi^\star_h(s_h,a_h)\right\|_{(W^{(n)}_{h,\phi^\star})^{-1}}\right] + 15\alpha_n\sqrt{\frac{dK}{n}}.
\end{aligned}
$$

Similarly, we obtain

$$
\begin{aligned}
V^{\pi_n}_{P^\star,f^{(n)}} &= \sum_{h=1}^H \mathbb{E}_{\substack{s_h\sim(P^\star,\pi_n) \\ a_h\sim\pi_n}}\left[f^{(n)}_h(s_h,a_h)\right] \\
&\leq \sum_{h=2}^H \mathbb{E}_{\substack{s_{h-1}\sim(P^\star,\pi_n) \\ a_{h-1}\sim\pi_n}}\left[\alpha_n\left\|\phi^\star_{h-1}(s_{h-1},a_{h-1})\right\|_{(U^{(n)}_{h-1,\phi^\star})^{-1}}\right] + \sqrt{K\zeta_n} \\
&\leq \sum_{h=1}^H \mathbb{E}_{\substack{s_h\sim(P^\star,\pi_n) \\ a_h\sim\pi_n}}\left[\alpha_n\left\|\phi^\star_h(s_h,a_h)\right\|_{(U^{(n)}_{h,\phi^\star})^{-1}}\right] + \sqrt{K\zeta_n}.
\end{aligned}
$$

Then, taking the summation of $V^{\pi_n}_{P^\star, \hat{b}^{(n)} + f^{(n)}}$ over $n \in [N]$, we have

$$
\sum_{n \in [N]} V^{\pi_n}_{P^\star, f^{(n)} + \hat{b}^{(n)}} + \sqrt{K \zeta_n}
$$

$$
\leq \sum_{n \in [N]} 15 \alpha_n \sqrt{\frac{dK}{n}} + 2 \sum_{n \in [N]} \sqrt{K \zeta_n} + \sum_{n \in [N]} \sum_{h=1}^{H} \mathop{\mathbb{E}}_{\substack{s_h \sim (P^\star, \pi_n) \\ a_h \sim \pi_n}} \left[ \gamma_n \left\| \phi_h^\star(s_h, a_h) \right\|_{(W_{h, \phi^\star}^{(n)})^{-1}} \right]
$$

$$
+ \sum_{n \in [N]} \sum_{h=1}^{H} \mathop{\mathbb{E}}_{\substack{s_h \sim (P^\star, \pi_n) \\ a_h \sim \pi_n}} \left[ \alpha_n \left\| \phi_h^\star(s_h, a_h) \right\|_{(U_{h, \phi^\star}^{(n)})^{-1}} \right]
$$

$$
\overset{(i)}{\leq} 17 \alpha_N \sqrt{dKN} + \gamma_N \sum_{h=1}^{H} \sqrt{N \sum_{n \in [N]} \mathop{\mathbb{E}}_{\substack{s_h \sim (P^\star, \pi_n) \\ a_h \sim \pi_n}} \left[ \left\| \phi_h^\star(s_h, a_h) \right\|_{(W_{h, \phi^\star}^{(n)})^{-1}}^2 \right]}
$$

$$
+ \alpha_N \sum_{h=1}^{H} \sqrt{KN \sum_{n \in [N]} \mathop{\mathbb{E}}_{\substack{s_h \sim (P^\star, \pi_n) \\ a_h \sim \mathcal{U}}} \left[ \left\| \phi_h^\star(s_h, a_h) \right\|_{(U_{h, \phi^\star}^{(n)})^{-1}}^2 \right]}
$$

$$
\overset{(ii)}{\leq} 17 \sqrt{\zeta} \sqrt{2 \beta_3 dK(K + d^2) N} + H \sqrt{45 \beta_3 \zeta dK(K + d^2)} \sqrt{dN \zeta}
$$

$$
+ H \sqrt{\beta_3 \zeta (K + d^2)} \sqrt{dKN\zeta}
$$

$$
\leq 32 \zeta H d \sqrt{\beta_3 K(d^2 + K) N},
$$

where $(i)$ follows from Cauchy-Schwarz inequality and importance sampling, and $(ii)$ follows from Lemma 10. Hence, the statement of Proposition 5 is verified. $\qquad\square$

## A.4 PROOF OF PROPOSITION 3

Based on Proposition 5, we argue that with enough number of iterations, RAFFLE can find $\hat{P}^\epsilon$ satisfying the condition in line 15 of Algorithm 1.

**Proposition 6** (Restatement of Proposition 3). *Fix any $\delta \in (0, 1), \epsilon > 0$. Suppose the algorithm runs for $N = \frac{2^{14} \beta_3 H^2 d^2 K(d^2 + K) \log^2(2|\Phi||\Psi|H^3 d^2 K(d^2 + K)/(\delta \epsilon^2))}{\epsilon^2}$ iterations, with probability at least $1 - \delta$, RAFFLE can find an $n_\epsilon \leq N$ in the exploration phase such that $2 \hat{V}^{\pi_{n_\epsilon}}_{\hat{P}^{(n_\epsilon)}, \hat{b}^{(n_\epsilon)}} + 2 \sqrt{K \zeta_{n_\epsilon}} \leq \epsilon$. In other words, Algorithm 1 can output $\hat{P}^\epsilon = \hat{P}^{(n_\epsilon)}$ satisfying the condition in line 15. In addition*

$$
\left| V^\pi_{P^\star, r} - V^\pi_{\hat{P}^{(n_\epsilon)}, r} \right| \leq \epsilon/2.
$$

*Proof.* We show that the algorithm terminates by contradiction. If it does not stop, applying Proposition 5, we have

$$
\epsilon N / 2 < \sum_{n \in [N]} \left( \hat{V}^{\pi_n}_{\hat{P}^{(n)}, \hat{b}^{(n)}} + \sqrt{K \zeta_n} \right)
$$

$$
\leq 32 \zeta H d \sqrt{\beta_3 K(d^2 + K) N}.
$$

Therefore,

$$
N < \frac{2^{12} \beta_3 H^2 d^2 K(d^2 + K) \zeta^2}{\epsilon^2}
$$

Recall $\zeta = N \zeta_N = \log(2|\Phi||\Psi|NH/\delta)$. Using the fact that $n \leq c \log^2(\alpha_n n) \Rightarrow n \leq 4c \log^2(\alpha_n c), \forall c \geq e^2, n \geq 1, \alpha_n \in \mathbb{R}^+$, it can be concluded that

$$
N < \frac{2^{14} \beta_3 H^2 d^2 K(d^2 + K) \log^2(2|\Phi||\Psi|H^3 d^2 K(d^2 + K)/(\delta \epsilon^2))}{\epsilon^2},
$$

which is a contradiction.

Therefore, there exists an $n_\epsilon = O(\frac{H^2 d^2 K (d^2 + K) \log^2(|\Phi||\Psi| H^3 d^2 K (d^2 + K)/(\delta \epsilon^2))}{\epsilon^2})$ such that $\hat{P}^\epsilon = \hat{P}^{(n_\epsilon)}$ satisfies

$$2\hat{V}_{\hat{P}^{(n_\epsilon)}, \hat{b}^{(n_\epsilon)}}^{\pi_{n_\epsilon}} + 2\sqrt{K \zeta_{n_\epsilon}} \leq \epsilon.$$

Combining Proposition 4, we finish the proof.

$\square$

*Proof of Theorem 2.* Recall that $\hat{P}^\epsilon$ is the output of RAFFLE in the $n_\epsilon$-iteration. Then, by Proposition 6

$$
\begin{aligned}
V_{P^\star, r}^\star &- V_{P^\star, r}^{\bar{\pi}} \\
&\leq V_{\hat{P}^\epsilon, r}^{\pi^\star} - V_{P^\star, r}^{\bar{\pi}} + \epsilon/2 \\
&\overset{(i)}{\leq} V_{\hat{P}^\epsilon, r}^{\bar{\pi}} - V_{P^\star, r}^{\bar{\pi}} + \epsilon/2 \\
&\leq \epsilon/2 + \epsilon/2 \\
&= \epsilon,
\end{aligned}
$$

where $(i)$ follows from the definition of $\bar{\pi}$. The number of trajectories $n_\epsilon H$ is at at most

$$O\left( \frac{H^3 d^2 K (d^2 + K) \log^2(|\Phi||\Psi| H^3 d^2 K (d^2 + K)/(\delta \epsilon^2))}{\epsilon^2} \right)$$

$\square$

# B  PROOF OF THEOREM 3

In this section, we adopt the same notations as in Appendix A. The following lemma provides an upper bound for the estimation error of any learned model from the true model.

**Lemma 4.** *Fix $\delta \in (0, 1)$, for any $h \in [H], n \in \mathbb{N}^+$, any policy $\pi$, with probability at least $1 - \delta/2$,*

$$\mathop{\mathbb{E}}_{\substack{s_h \sim (P^\star, \pi) \\ s_h \sim \pi}} \left[ f_h^{(n)}(s_h, a_h) \right] \leq 2\sqrt{K \zeta_n} + 2\hat{V}_{\hat{P}^{(n)}, \hat{b}^{(n)}}^{\pi_n}.$$

*Proof.* Recall that $f_h^{(n)}(s, a) = \left\| \hat{P}_h^{(n)}(\cdot|s, a) - P_h^\star(\cdot|s, a) \right\|_{TV}$. Fix any policy $\pi$, for any $h \geq 2$, we have

$$
\begin{aligned}
\mathop{\mathbb{E}}_{\substack{s_h \sim (\hat{P}^{(n)}, \pi) \\ a_h \sim \pi}} \left[ \hat{Q}_{h, \hat{P}^{(n)}, \hat{b}^{(n)}}^{\pi}(s_h, a_h) \right] &= \mathop{\mathbb{E}}_{\substack{s_{h-1} \sim (\hat{P}^{(n)}, \pi) \\ a_{h-1} \sim \pi}} \left[ \hat{P}_h^{(n)} \hat{V}_{h, \hat{P}^{(n)}, \hat{b}^{(n)}}^{\pi}(s_{h-1}, a_{h-1}) \right] \\
&\leq \mathop{\mathbb{E}}_{\substack{s_{h-1} \sim (\hat{P}^{(n)}, \pi) \\ a_{h-1} \sim \pi}} \left[ \min\left\{ 1, \hat{b}_{h-1}^{(n)}(s_{h-1}, a_{h-1}) + \hat{P}_{h-1}^{(n)} \hat{V}_{h, \hat{P}^{(n)}, \hat{b}^{(n)}}^{\pi}(s_{h-1}, a_{h-1}) \right\} \right] \\
&= \mathop{\mathbb{E}}_{\substack{s_{h-1} \sim (\hat{P}^{(n)}, \pi) \\ a_{h-1} \sim \pi}} \left[ \hat{Q}_{h-1, \hat{P}^{(n)}, \hat{b}^{(n)}}^{\pi}(s_{h-1}, a_{h-1}) \right] \\
&\leq \dots \\
&\leq \mathop{\mathbb{E}}_{a_1 \sim \pi} \left[ \hat{Q}_{1, \hat{P}^{(n)}, \hat{b}^{(n)}}^{\pi}(s_1, a_1) \right] \\
&= \hat{V}_{\hat{P}^{(n)}, \hat{b}^{(n)}}^{\pi}. \tag{19}
\end{aligned}
$$

Hence, for $h \geq 2$, we have

$$
\mathbb{E}_{\substack{s_h \sim (\hat{P}^{(n)}, \pi) \\ a_h \sim \pi}} \left[ f_h^{(n)}(s_h, a_h) \right] \overset{(i)}{\leq} \mathbb{E}_{\substack{s_{h-1} \sim (\hat{P}^{(n)}, \pi) \\ a_{h-1} \sim \pi}} \left[ \hat{b}_{h-1}^{(n)}(s_{h-1}, a_{h-1}) \right]
$$

$$
\overset{(ii)}{\leq} \mathbb{E}_{\substack{s_{h-1} \sim (\hat{P}^{(n)}, \pi) \\ a_{h-1} \sim \pi}} \left[ \hat{Q}_{h-1, \hat{P}^{(n)}, \hat{b}^{(n)}}^{\pi}(s_{h-1}, a_{h-1}) \right]
$$

$$
\overset{(iii)}{\leq} \hat{V}_{\hat{P}^{(n)}, \hat{b}^{(n)}}^{\pi}, \tag{20}
$$

where $(i)$ follows from Equation (18), $(ii)$ follows from the definition of $\hat{Q}_{h-1, \hat{P}^{(n)}, \hat{b}^{(n)}}^{\pi}(s_{h-1}, a_{h-1})$ and $(iii)$ follows from Equation (19).

$$
\mathbb{E}_{\substack{\sim (P^\star, \pi) \\ s_h \sim \pi}} \left[ f_h^{(n)}(s_h, a_h) \right] \leq \mathbb{E}_{\substack{s_h \sim (\hat{P}^{(n)}, \pi) \\ a_h \sim \pi}} \left[ f_h^{(n)}(s_h, a_h) \right] + \left| \mathbb{E}_{\substack{s_h \sim (P^\star, \pi) \\ a_h \sim \pi}} \left[ f_h^{(n)}(s_h, a_h) \right] - \mathbb{E}_{\substack{s_h \sim (\hat{P}^{(n)}, \pi) \\ a_h \sim \pi}} \left[ f_h^{(n)}(s_h, a_h) \right] \right|
$$

$$
\overset{(i)}{\leq} (\hat{V}_{\hat{P}^{(n)}, \hat{b}^{(n)}}^{\pi} + \sqrt{K\zeta_n}) + \left( \sqrt{K\zeta_n} + \hat{V}_{\hat{P}^{(n)}, \hat{b}^{(n)}}^{\pi} \right)
$$

$$
\overset{(ii)}{\leq} 2\sqrt{K\zeta_n} + 2\hat{V}_{\hat{P}^{(n)}, \hat{b}^{(n)}}^{\pi_n},
$$

where the first term in $(i)$ is due to Equation (20) and Equation (14), the second term in $(i)$ is due to Proposition 4 and $(ii)$ follows from the definition of $\pi_n$. $\qquad \square$

*Proof of Theorem 3.* By Proposition 6, let $n_\epsilon = O\left( \frac{H^2 d^2 K(d^2+K) \log^2(|\Phi||\Psi| N H^3 d^2 K(K+d^2)/(\delta\epsilon^2))}{\epsilon^2} \right)$, with no more than $n_\epsilon H$ trajectories, RAFFLE can learn a model $\hat{P}^\epsilon$, bonus $\hat{b}^\epsilon$ and policy $\pi_\epsilon$ at the $n_\epsilon$-th iteration satisfying $2V_{\hat{P}^\epsilon, \hat{b}^\epsilon}^{\pi_\epsilon} + 2\sqrt{K\zeta_{n_\epsilon}} \leq \epsilon$. Then, following Lemma 4, we have

$$
\mathbb{E}_{\substack{s_h \sim (P^\star, \pi) \\ s_h \sim \pi}} \left[ \|\hat{P}_h^\epsilon(\cdot|s_h, a_h) - P_h^\star(\cdot|s_h, a_h)\|_{TV} \right] \leq 2\sqrt{K\zeta_{n_\epsilon}} + 2V_{\hat{P}^\epsilon, \hat{b}^\epsilon}^{\pi_\epsilon} \leq \epsilon.
$$

$\qquad \square$

# C    Proof of Theorem 1 (Lower Bound on Sample Complexity)

## C.1    Step 1: Construction of Hard MDP instances

Our hard MDP instances is inspired by Domingues et al. (2020) for tabular MDPs. However, the lower bound in Domingues et al. (2020) requires $S \geq K$, where $S, K$ denote the cardinality of state and action space respectively. Our hard MDP instances remove the assumption that $S \geq K$ by constructing the action set with two types of actions. The first type of actions is mainly used to form a large state space through a tree structure. The second type of actions is mainly used to distinguish different MDPs. Such a construction allows us to separately treat the state space and the action space, so that both state and action spaces can be arbitrarily large. We then explicitly define the feature vectors for all state-action pairs and show our hard MDP instances have a low-rank structure with dimension $d = S$. In a nutshell, we construct a family of $HdK$ MDPs that are hard to distinguish in KL divergence, while the corresponding optimal policies are very different as shown in Figure 1.

First, we define a reference MDP $\mathcal{M}_0$ as follows. We start with the construction of the state space $\mathcal{S}$ and the action space $\mathcal{A}$.

- Let $\mathcal{S} = \{s_w, s_o, s_g, s_b\} \bigcup_{i \in [D], j \in [2^{i-1}]} \{s^{i,j}\}$, where $s_w, s_o, s_g$ and $s_b$ denote 'waiting state', 'outlier state', 'good state', and 'bad state', respectively. The states in $\bigcup_{i \in [D], j \in [2^{i-1}]} \{s^{i,j}\}$ form a binary tree, where $s^{i,j}$ denotes the $j$-th branch node of the layer $i$.

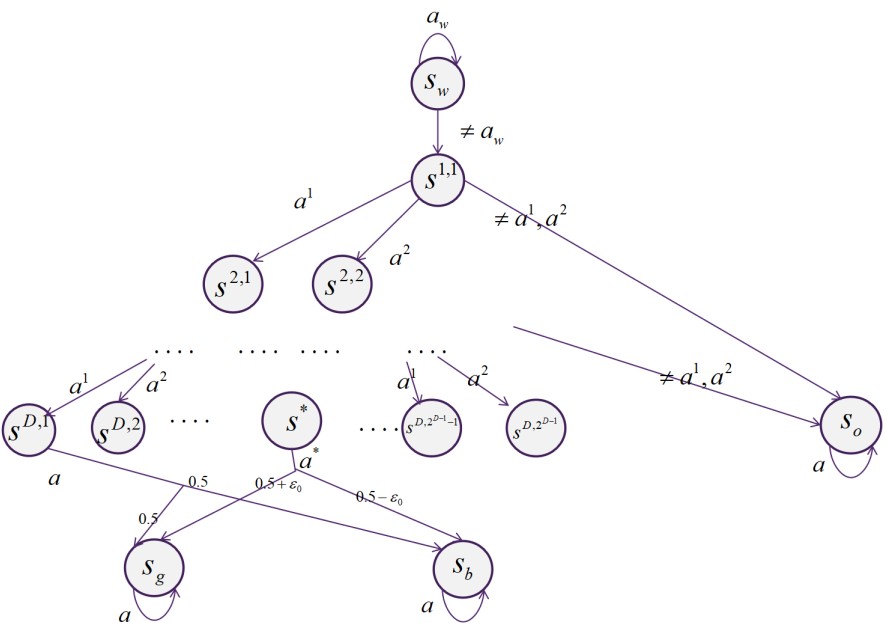

Figure 1: Hard MDP instances.

- Let $\mathcal{A} = \{a_w, a^1, a^2\} \bigcup \mathcal{A}_0$, where $a_w$ denotes 'waiting action', $a^1, a^2$ are two unique actions that form the binary tree, and $|\mathcal{A}_0| = K - 3$.

Then, the transition probabilities of $\mathcal{M}_0$ are specified through the following rules.

- The initial state is the waiting state $s_w$.

- If the agent takes the waiting action $a_w$ before time step $\bar{H}$, waiting state $s_w$ stays on itself. Otherwise, $s_w$ transits to the root state $s^{1,1}$ of the binary tree.

  Mathematically, $\mathbb{P}_h[s_w|s_w, a_w] = \mathbb{1}_{h \leq \bar{H}}$, and $\mathbb{P}_h[s^{1,1}|s_w, a] = \mathbb{1}_{a \neq a_w \text{ or } h > \bar{H}}$.

- When $i < D$, for states $s^{i,j}$ in the binary tree, we have the following transition rules:

  - If the agent takes actions $a_1$ or $a_2$, $s^{i,j}$ deterministically transits to its children $s^{i+1,2j-1}$ or $s^{i+1,2j}$, respectively.
    Mathematically, $\mathbb{P}_h[s^{i+1,2j-1}|s^{i,j}, a^1] = 1$, and $\mathbb{P}[s^{i+1,2j}|s^{i,j}, a^2] = 1$.

  - If the agent takes any action other than $a^1$ or $a^2$, the agent will reach the outlier state $s_o$.
    Mathematically, $\mathbb{P}_h[s_o|s^{i,j}, a] = 1, \forall a \neq a^1, a^2$.

- Leaf state $s^{D,j}$ uniformly transits to good state $s_g$ and bad state $s_b$ no matter what action the agent takes.

  Mathematically, $\mathbb{P}_h[s_g|s^{D,j}, a] = \mathbb{P}_h[s_b|s^{D,j}, a] = \frac{1}{2}, \forall a \in \mathcal{A}$.

- Good state $s_g$, bad state $s_b$, and outlier state $s_o$ are absorbing states.

Now, we define the features as follows, which are $S$-dimensional vectors.

$s_w$      $\phi_h(s_w, a_w) = (1, 0, \mathbf{0}_{S-5}, 0, 0, 0)$      $\mu_h(s_w) = (\mathbb{1}_{h \leq \bar{H}}, 0, \mathbf{0}_{S-5}, 0, 0, 0)$
$\phi_h(s_w, a) = (0, 1, \mathbf{0}_{S-5}, 0, 0, 0), a \neq a_w$      $\mu_h(s^{1,1}) = (\mathbb{1}_{h > \bar{H}}, 1, \mathbf{0}_{S-5}, 0, 0, 0)$

$s^{i,j}, i < D$    $\phi_h(s^{i,j}, a^\omega) = (0, 0, \mathbf{e}_{i+1, 2j+\omega-2}, 0, 0, 0), \omega = 1, 2$    $\mu_h(s^{k,\ell}) = (0, 0, \mathbf{e}_{k,\ell}, 0, 0, 0), 1 < k \leq D$
$\phi_h(s^{i,j}, a) = (0, 0, \mathbf{0}_{S-5}, 1, 0, 0), a \neq a^1, a^2$    $\mu_h(s_o) = (0, 0, \mathbf{0}_{S-5}, 1, 0, 0)$

$s^{D,j}$      $\phi_h(s^{D,j}, a) = (0, 0, \mathbf{0}_{S-5}, 0, \frac{1}{2}, \frac{1}{2})$      $\mu_h(s_g) = (0, 0, \mathbf{0}_{S-5}, 0, 1, 0)$

$s_g, s_b$      $\phi_h(s_g, a) = \mu_h(s_g), \quad \phi_h(s_b, a) = \mu_h(s_b)$      $\mu_h(s_b) = (0, 0, \mathbf{0}_{S-5}, 0, 0, 1)$
$s_o$      $\phi_h(s_o, a) = \mu_h(s_o),$

where $\mathbf{0}_{S-5} \in \mathbb{R}^{S-5}$ denotes the $S - 5$ dimension vector with all zeros and $\mathbf{e}_{i,j} \in \mathbb{R}^{S-5}$ denotes the one-hot vector that is zero everywhere except the $(2^{i-1} + j - 2)$-th coordinate. Here $2 \leq i \leq D, 1 \leq j \leq 2^{i-1}$.

For each $(h^*, \ell^*, a^*) \in \{1 + D, \ldots, \bar{H} + D\} \times [2^{D-1}] \times \mathcal{A}$, we define an MDP $\mathcal{M}_{(h^*, \ell^*, a^*)}$ through $\mathcal{M}_0$. Specifically, the only difference of $\mathcal{M}_{(h^*, \ell^*, a^*)}$ from $\mathcal{M}_0$ is that the transition probability from the leaf state $s^{D,\ell^*}$ and action $a^*$ to the good state $s_g$ increases $\epsilon_0$, i.e. $\mathbb{P}_{h^*}[s_g | s^{D,\ell^*}, a^*] = \frac{1}{2} + \epsilon_0$, and $\mathbb{P}_{h^*}[s_b | s^{D,\ell^*}, a^*] = \frac{1}{2} - \epsilon_0$, where $\epsilon_0$ will be specified later. We note that the features of $\mathcal{M}_{(h^*, \ell^*, a^*)}$ are the same as those of $\mathcal{M}_0$ except that $\phi_{h^*}(s^{D,\ell^*}, a^*) = (0, 0, \mathbf{0}_{S-5}, 0, \frac{1}{2} + \epsilon_0, \frac{1}{2} - \epsilon_0)$.

We remark here that the cardinality $K$ of $\mathcal{A}$ can be arbitrarily large, so that the resulting lower bound will hold for both $d \leq K$ and $d \geq K$ regimes. In addition, although in our hard instances, $d = S$, it is straightforward to generalize it to the regime with $S > d$ if we set the outlier state $S_o$ to be a set of outlier states $\mathcal{S}_o$.

**Definition of reward**: the reward can only be attained in two special states: the good state $s_g$ and the outlier state $s_o$ at the last stage $H$.

$$\forall \mathcal{S} \in \mathcal{A}, a \in \mathcal{A}, r_h(s, a) = \mathbb{1}_{\{s=s_g, h=H\}} + \frac{1}{2} \mathbb{1}_{\{s=s_o, h=H\}},$$

and $r_h(s, a)$ still belongs to $[0, 1]$.

### C.2    STEP 2: ANALYSIS OF HARD MDP INSTANCES

*Proof of Theorem 1.* Let $\epsilon_0 = 2\epsilon$.

For any MDP $\mathcal{M}_{h^*, a^*}$, the optimal policy is to take action $a_w$ to stay at state $s_w$ until stage $h^* - D$, and then take the corresponding action to the only state $s^{D,\ell^*}$ at stage $h^*$. At state $s^{D,\ell^*}$, the agent takes the only optimal action $a^*$. The optimal value function $V^*_{\mathcal{M}_{(h^*, \ell^*, a^*)}} = \frac{1}{2} + \epsilon_0$, and the value function of the output policy of Alg is given by

$$V^{\hat{\pi}_\tau}_{\mathcal{M}_{(h^*, \ell^*, a^*)}} = \frac{1}{2} + \epsilon_0 \mathbb{P}^{\hat{\pi}_\tau}_{\mathcal{M}_{(h^*, \ell^*, a^*)}}[s_{h^*} = s^{\ell^*}, a_{h^*} = a^*], \tag{21}$$

where $\mathbb{P}^{\hat{\pi}_\tau}_{\mathcal{M}_{(h^*, \ell^*, a^*)}}$ is the probability distribution over the states and actions $(s_h, a_h)$ following the Markov policy $\hat{\pi}_\tau$ in the MDP $\mathcal{M}_{(h^*, \ell^*, a^*)}$. We remark that the reward of outlier state are specially designed to be $1/2$ at stage $H$ to make Equation (21) hold for policy $\hat{\pi}_\tau$ falling into the outlier state.

Hence,

$$V^*_{\mathcal{M}_{(h^*, \ell^*, a^*)}} - V^{\hat{\pi}_\tau}_{\mathcal{M}_{(h^*, \ell^*, a^*)}} = \epsilon_0(1 - \mathbb{P}^{\hat{\pi}_\tau}_{\mathcal{M}_{(h^*, \ell^*, a^*)}}[s_{h^*} = s^{\ell^*}, a_{h^*} = a^*])$$

$$= 2\epsilon(1 - \mathbb{P}^{\hat{\pi}_\tau}_{\mathcal{M}_{(h^*, \ell^*, a^*)}}[s_{h^*} = s^{\ell^*}, a_{h^*} = a^*]).$$

and

$$V^*_{\mathcal{M}_{(h^*, \ell^*, a^*)}} - V^{\hat{\pi}_\tau}_{\mathcal{M}_{(h^*, \ell^*, a^*)}} \leq \epsilon \Leftrightarrow \mathbb{P}^{\hat{\pi}_\tau}_{\mathcal{M}_{(h^*, \ell^*, a^*)}}[s_{h^*} = s^{\ell^*}, a_{h^*} = a^*] \geq \frac{1}{2}. \tag{22}$$

The transitions of all MDPs are the same when the leaves states are reached. We define the event

$$\varepsilon^\tau_{(h^*, \ell^*, a^*)} = \left\{ \mathbb{P}^{\hat{\pi}_\tau}_{\mathcal{M}_{(h^*, \ell^*, a^*)}}[s_{h^*} = s^{\ell^*}, a_{h^*} = a^*] \geq \frac{1}{2} \right\}.$$

From Equation (22), the event is equal to the event $\{V^*_{\mathcal{M}_{(h^*,\ell^*,a^*)}} - V^{\hat{\pi}_\tau}_{\mathcal{M}_{(h^*,\ell^*,a^*)}} \leq \epsilon\}$. As a result,

$$\mathbb{P}_{(h^*,\ell^*,a^*)}\left[\varepsilon^\tau_{(h^*,\ell^*,a^*)}\right] = \mathbb{P}_{(h^*,\ell^*,a^*)}\left[V^*_{\mathcal{M}_{(h^*,\ell^*,a^*)}} - V^{\hat{\pi}_\tau}_{\mathcal{M}_{(h^*,\ell^*,a^*)}} \leq \epsilon\right] \geq 1 - \delta.$$

Recall that $N^\tau_{(h^*,\ell^*,a^*)} = \sum_{n=1}^\tau \mathbb{1}_{\{(s^n_{h^*},s^n_{h^*})=(s\ell^*,a^*)\}}$ such that $\sum_{(h^*,\ell^*,a^*)} N^\tau_{(h^*,\ell^*,a^*)} \leq \tau$. This inequality holds because the agent is likely to fall into the outlier state $s_o$. We denote $\mathbb{P}_0$ and $\mathbb{E}_0$ to be with respect to $\mathcal{M}_0$.

Now, we invoke an intermediate result in the proof of Theorem 7 in Domingues et al. (2020) to conclude that

$$\mathbb{E}_0\left[N^\tau_{(h^*,\ell^*,a^*)}\right] \geq \frac{1}{16\epsilon^2}\left[\left(1 - \mathbb{P}_0\left[\{\varepsilon^\tau_{(h^*,\ell^*,a^*)}\}\right]\right)\log\left(\frac{1}{\delta}\right) - \log(2)\right].$$

Summing over all $(h^*,\ell^*,a^*)$, we have

$$\mathbb{E}_0[\tau] \geq \sum_{(h^*,\ell^*,a^*)} \mathbb{E}_0\left[N^\tau_{(h^*,\ell^*,a^*)}\right]$$

$$\geq \frac{1}{16\epsilon^2}\left[\left(\bar{H}LK - \sum_{(h^*,\ell^*,a^*)} \mathbb{P}_0\left[\varepsilon^\tau_{(h^*,\ell^*,a^*)}\right]\right)\log(\frac{1}{\delta}) - \bar{H}LK\log 2\right]. \qquad (23)$$

Notice that

$$\sum_{(h^*,\ell^*,a^*)} \mathbb{P}_0\left[\varepsilon^\tau_{(h^*,\ell^*,a^*)}\right] = \mathbb{E}_0\left[\sum_{(h^*,\ell^*,a^*)} \mathbb{1}_{\{\mathbb{P}^{\hat{\pi}_\tau}_{\mathcal{M}_{(h^*,\ell^*,a^*)}}[s_{h^*}=s\ell^*,a_{h^*}=a^*]\geq\frac{1}{2}\}}\right] \leq 1. \qquad (24)$$

Substituting Equation (24) into Equation (23) yields

$$\mathbb{E}_0[\tau] \geq \frac{1}{16\epsilon^2}\left[(\bar{H}LK - 1)\log(\frac{1}{\delta}) - \bar{H}LK\log 2\right]$$

$$\geq \frac{1}{32\epsilon^2}\bar{H}LK\log(\frac{1}{\delta}),$$

where we use the fact that $\delta < 1/16$. With the assumption of $K \geq 3, S \geq 6$, we have $d = S$. Taking $\bar{H} = \frac{H}{3}$ and with the assumption of $D \leq H/3$, we have

$$\mathbb{E}_0[\tau] = \Omega\left(\frac{HdK}{\epsilon^2}\log(\frac{1}{\delta})\right).$$

Then following the analysis similar to that for Corollary 8 in Domingues et al. (2020), with probability at least $1 - \delta$, the number of iterarion is at least

$$\Omega\left(\frac{HdK}{\epsilon^2}\log(\frac{1}{\delta})\right).$$

$\square$

# D  ALGORITHM 2: REPLEARN AND PROOF OF THEOREM 4

We first present the full algorithm in Section 6 below as Algorithm 2.

## D.1  SUPPORTING LEMMAS

We first show that $Q^\pi_{\hat{P},h,r}$ can approximate $Q^\pi_{P^*,h,r}$ well over distribution $\{q_h\}_{h\in[H]}$ by following two lemmas.

**Lemma 5.** *Given any $\delta \in (0,1)$. Let $\hat{P}$ be the output of Algorithm 1, for any policy $\pi$ and rewards $r$, with probability at least $1 - \delta$, we have*

$$\mathbb{E}_{(s_h,a_h)\sim(\hat{P},\pi)}\left[\left|Q^\pi_{P^*,h,r}(s_h,a_h) - Q^\pi_{\hat{P},h,r}(s_h,a_h)\right|\right] \leq \epsilon$$

$$\mathbb{E}_{(s_h,a_h)\sim(P^*,\pi)}\left[\left|Q^\pi_{P^*,h,r}(s_h,a_h) - Q^\pi_{\hat{P},h,r}(s_h,a_h)\right|\right] \leq \epsilon.$$

---

**Algorithm 2 RepLearn**: Representation Learning in Planning Phase of RAFFLE

---

1: **Input:** Sample size $N_f$, state-action pair distributions $\{q_h\}_{h=1}^H$, special designed reward function $\{r^{h,t}\}_{h\in[H],t\in[T]}$ and policy $\{\pi^t\}_{t\in[T]}$, and estimated transition kernel $\hat{P}$ from the output of Algorithm 1, model class: $\Phi$.
2: Initialize $\mathcal{D}_h^{0,0} = \emptyset$.
3: **for** $n = 1, \ldots, N_f$ **do**
4:    **for** $h = 1, \ldots, H$ **do**
5:       **for** $t = 1, \ldots, T$ **do**
6:          Choose $(s_h^{n,t}, a_h^{n,t}) \sim q_h$ and add $(s_h^{n,t}, a_h^{n,t})$ to dataset $\mathcal{D}_h^{n,t} = \mathcal{D}_h^{n-1,t} \cup (s_h^{n,t}, a_h^{n,t})$.
7:       **end for**
8:    **end for**
9: **end for**
10: **for** $h = 1, \ldots, H$ **do**
11:    Learn $(\tilde{\phi}_h, \tilde{w}_h^1, \ldots, \tilde{w}_h^T)$ as in Equation (4).
12: **end for**
13: **Output:** $\tilde{\phi} = \{\tilde{\phi}_h\}_{h\in[H]}$.

---

*Proof.* Define $\hat{f}_h(s_h, a_h) = \left\|\hat{P}_h(\cdot|s_h, a_h) - P^\star(\cdot|s_h, a_h)\right\|_{TV}$ and $\hat{f}$ is a collection of all $\hat{f}_h$, i.e. $\left\{\hat{f}_h\right\}_{h\in[H]}$.

$$
\begin{aligned}
\mathbb{E}_{(s_h,a_h)\sim(\hat{P},\pi)} &\left[\left|Q_{P^\star,h,r}^\pi(s_h,a_h) - Q_{\hat{P},h,r}^\pi(s_h,a_h)\right|\right] \\
&\overset{(i)}{\leq} \mathbb{E}_{(s_h,a_h)\sim(\hat{P},\pi)}\left[\hat{Q}_{h,\hat{P},\hat{f}}^\pi(s_h,a_h)\right] \\
&\overset{(ii)}{\leq} \hat{V}_{\hat{P},\hat{f}}^\pi \\
&\overset{(iii)}{\leq} \epsilon/2,
\end{aligned}
\tag{25}
$$

where $(i)$ follows from Equation (15), $(ii)$ follows from the definition of $\hat{V}$ and $\hat{Q}$, and $(iii)$ follows from the proof of Theorem 2.

Then for the second inequality,

$$
\begin{aligned}
\mathbb{E}_{(s_h,a_h)\sim(P^\star,\pi)} &\left[\left|Q_{P^\star,h,r}^\pi(s_h,a_h) - Q_{\hat{P},h,r}^\pi(s_h,a_h)\right|\right] \\
\leq\ & \mathbb{E}_{(s_h,a_h)\sim(\hat{P},\pi)}\left[\left|Q_{P^\star,h,r}^\pi(s_h,a_h) - Q_{\hat{P},h,r}^\pi(s_h,a_h)\right|\right] \\
& + \left|\mathbb{E}_{(s_h,a_h)\sim(P^\star,\pi)}\left[\left|Q_{P^\star,h,r}^\pi(s_h,a_h) - Q_{\hat{P},h,r}^\pi(s_h,a_h)\right|\right] - \mathbb{E}_{(s_h,a_h)\sim(\hat{P},\pi)}\left[\left|Q_{P^\star,h,r}^\pi(s_h,a_h) - Q_{\hat{P},h,r}^\pi(s_h,a_h)\right|\right]\right| \\
\overset{(i)}{\leq}\ & \epsilon/2 + V_{\hat{P},\hat{f}}^\pi \\
\overset{(ii)}{\leq}\ & \epsilon,
\end{aligned}
$$

where the first term in $(i)$ is due to Equation (25) and Equation (14), the second term in $(i)$ is due to Step 1 in Proposition 4 and $(ii)$ follows from the proof of Theorem 2. $\qquad\square$

**Lemma 6.** *Given any $\delta, \epsilon \in (0,1)$ and the output of Algorithm 1, under Assumption 2, for any policy $\pi$ and rewards $r$, let the input distributions $\{q_h\}_{h=1}^H$ are bounded with constant $C_B$. Denote $C_{\min} = \frac{C_B}{\eta_{\min}}$. Then with probability at least $1 - \delta$, for each $h \in [H]$,*
$$
\mathbb{E}_{(s_h,a_h)\sim q_h}\left[\left|Q_{P^\star,h,r}^\pi(s_h,a_h) - Q_{\hat{P},h,r}^\pi(s_h,a_h)\right|\right] \leq \epsilon C_{\min}.
$$

*Proof.* First, together with Assumption 2, for any $(s,a) \in \mathcal{S} \times \mathcal{A}$, we have $q_h(s,a) \leq C_{\min} \mathbb{P}_h^{\pi^0}(s,a)$, then

$$\mathbb{E}_{(s_h,a_h) \sim q_h} \left[ \left| Q_{P^\star,h,r}^\pi(s_h,a_h) - Q_{\hat{P},h,r}^\pi(s_h,a_h) \right| \right]$$

$$= \int q_h(s_h,a_h) \left| Q_{P^\star,h,r}^\pi(s_h,a_h) - Q_{\hat{P},h,r}^\pi(s_h,a_h) \right| ds_h da_h$$

$$\overset{(i)}{=} \int C_{\min} \mathbb{P}_h^{\pi^0}(s_h) \left| Q_{P^\star,h,r}^\pi(s_h,a_h) - Q_{\hat{P},h,r}^\pi(s_h,a_h) \right| ds_h da_h$$

$$= C_{\min} \mathbb{E}_{(s_h,a_h) \sim (P^\star,\pi^0)} \left[ \left| Q_{P^\star,h,r}^\pi(s_h,a_h) - Q_{\hat{P},h,r}^\pi(s_h,a_h) \right| \right]$$

$$\overset{(ii)}{\leq} \epsilon C_{\min},$$

where $(i)$ follows Assumption 2 and $(ii)$ follows Lemma 5. $\qquad\square$

The lemma above shows that for any reward $r$, $Q_{\hat{P},h,r}^\pi(s_h,a_h)$ can be the target of $Q_{P^\star,h,r}^\pi(s_h,a_h)$ when $(s_h,a_h)$ are chosen from given distribution $q_h$.

We then show that for any $h$, when reward $r$ is set to be zero at step $h$, $Q_{P^\star,h,r}^\pi$ has a linear structure w.r.t $\phi_h^\star$.

**Lemma 7.** *For any $h \in [H]$, policy $\pi$ and given $(s_h,a_h) \in \mathcal{S} \times \mathcal{A}$, given any $r$ such that $r$ is set to be zero at step $h$, i.e. $r_h = 0$, then $Q_{P^\star,h,r}^\pi(s_h,a_h)$ is linear with respect to $\phi^\star(s_h,a_h)$, i.e. there exist a $w_h^\star$ such that $Q_{P^\star,h,r}^\pi(s_h,a_h) = \langle \phi_h^\star(s_h,a_h), w_h^\star \rangle$.*

*Proof.*

$$Q_{P^\star,h,r}^\pi(s_h,a_h) = r_h(s_h,a_h) + \mathbb{E}_{s_{h+1} \sim P^\star(\cdot|s_h,a_h)} \left[ V_{P^\star,h+1,r}^\pi(s_{h+1}) \middle| s_h,a_h \right]$$

$$= \int P^\star(s_{h+1}|s_h,a_h) V_{P^\star,h+1,r}^\pi(s_{h+1}) ds_{h+1}$$

$$= \left\langle \phi_h^\star(s_h,a_h), \int \mu_h^\star(s_{h+1}) V_{P^\star,h+1,r}^\pi(s_{h+1}) ds_{h+1} \right\rangle$$

$$= \langle \phi_h^\star(s_h,a_h), w_h^\star \rangle,$$

where $w_h^\star = \int \mu_h^\star(s_{h+1}) V_{P^\star,h+1,r}^\pi(s_{h+1}) ds_{h+1}$.

$\qquad\square$

## D.2 PROOF OF THEOREM 4

*Proof of Theorem 4.* We allow $\phi$ and $Q_{\hat{P},h,r^{h,t}}^{\pi^t}$ to apply to all the samples in a dataset matrix simultaneously, i.e. $\phi_h(\mathcal{D}_h^{N_f,t}) = (\phi_h(s_h^{1,t}), \ldots, \phi_h(s_h^{N_f,t}))^\top \in \mathbb{R}^{N_f \times d}$ and $Q_{\hat{P},h,r^{h,t}}^{\pi^t}(\mathcal{D}_h^{N_f,t}) = (Q_{\hat{P},h,r^{h,t}}^{\pi^t}(s_h^{1,t}), \ldots, Q_{\hat{P},h,r^{h,t}}^{\pi^t}(s_h^{N_f,t}))^\top \in \mathbb{R}^{N_f}$. After we estimating $\tilde{\phi}_h$ and then taking it as known, from the property of linear regression in Equation (4), for any reward function $r^{h,t}$, any policy $\pi^t$ we got

$$\widetilde{w}_h^t = (\tilde{\phi}_h(\mathcal{D}_h^{N_f,t})^\top \tilde{\phi}_h(\mathcal{D}_h^{N_f,t}))^\dagger \tilde{\phi}_h(\mathcal{D}_h^{N_f,t})^\top Q_{\hat{P},h,r^{h,t}}^\pi(\mathcal{D}_h^{N_f,t})$$

$$\tilde{\phi}_h(\mathcal{D}_h^{N_f,t})\widetilde{w}_h^t = P_{\tilde{\phi}_h(\mathcal{D}_h^{N_f,t})} Q_{\hat{P},h,r^{h,t}}^{\pi^t}(\mathcal{D}_h^{N_f,t}),$$

where $P_{\tilde{\phi}_h(\mathcal{D}_h^{N_f,t})} = \tilde{\phi}_h(\mathcal{D}_h^{N_f,t})(\tilde{\phi}_h(\mathcal{D}_h^{N_f,t})^\top \tilde{\phi}_h(\mathcal{D}_h^{N_f,t}))^\dagger \tilde{\phi}_h(\mathcal{D}_h^{N_f,t})^\top$, represents the projection operator to the column spaces of $\tilde{\phi}_h(\mathcal{D}_h^{N_f,t})$. Then

$$
\begin{aligned}
&\sum_{t\in[T]} \left\| P_{\tilde{\phi}_h(\mathcal{D}_h^{N_f,t})}^\perp Q_{\hat{P},h,r^{h,t}}^{\pi^t}(\mathcal{D}_h^{N_f,t}) \right\|^2 \\
&= \sum_{t\in[T]} \left\| \tilde{\phi}_h(\mathcal{D}_h^{N_f,t})\widetilde{w}_h^t - Q_{\hat{P},h,r^{h,t}}^{\pi^t}(\mathcal{D}_h^{N_f,t}) \right\|^2 \\
&\overset{(i)}{\leq} \sum_{t\in[T]} \left\| \phi_h^\star(\mathcal{D}_h^{N_f,t})w_h^{t\,\star} - Q_{\hat{P},h,r^{h,t}}^{\pi^t}(\mathcal{D}_h^{N_f,t}) \right\|^2 \\
&= \sum_{t\in[T]} \sum_{n=1}^{N_f} \left( Q_{P^\star,h,r^{h,t}}^{\pi^t}(s_h^{n,t},a_h^{n,t}) - Q_{\hat{P},h,r^{h,t}}^{\pi^t}(s_h^{n,t},a_h^{n,t}) \right)^2 \\
&\overset{(ii)}{\leq} N_f \sum_{t\in[T]} \mathbb{E}_{(s_h,a_h)\sim q_h}\left[ \left( Q_{P^\star,h,r^{h,t}}^{\pi^t}(s_h^{n,t},a_h^{n,t}) - Q_{\hat{P},h,r^{h,t}}^{\pi^t}(s_h^{n,t},a_h^{n,t}) \right)^2 \right] + \sqrt{\frac{TN_f\log\frac{2}{\delta}}{2}} \\
&\overset{(iii)}{\leq} N_f \sum_{t\in[T]} \mathbb{E}_{(s_h,a_h)\sim q_h}\left[ \left| Q_{P^\star,h,r^{h,t}}^{\pi^t}(s_h^{n,t},a_h^{n,t}) - Q_{\hat{P},h,r^{h,t}}^{\pi^t}(s_h^{n,t},a_h^{n,t}) \right| \right] + \sqrt{\frac{TN_f\log\frac{2}{\delta}}{2}} \\
&\overset{(iv)}{\leq} \epsilon C_{\min} N_f T + \sqrt{\frac{TN_f\log\frac{2}{\delta}}{2}},
\end{aligned}
\tag{26}
$$

where $(i)$ follows from minimality of $\{\tilde{\phi}_h^t\}_{t\in[T]}$ and $\{\widetilde{w}_h^t\}_{t\in[T]}$, $(ii)$ follows Hoeffding's inequality, $(iii)$ follows that $\left| Q_{P^\star,h,r^{h,t}}^{\pi^t}(s_h^{n,t},a_h^{n,t}) - Q_{\hat{P},h,r^{h,t}}^{\pi^t}(s_h^{n,t},a_h^{n,t}) \right| \leq 1$ and $(iv)$ follows Lemma 6.

As a result:

$$
\begin{aligned}
&\sum_{t\in[T]} \left\| P_{\tilde{\phi}_h(\mathcal{D}_h^{N_f,t})}^\perp \phi_h^\star(\mathcal{D}_h^{N_f,t})w_h^{t\,\star} \right\|^2 \\
&= \sum_{t\in[T]} \left\| P_{\tilde{\phi}_h(\mathcal{D}_h^{N_f,t})}^\perp Q_{P^\star,h,r^{h,t}}^{\pi^t}(\mathcal{D}_h^{N_f,t}) \right\|^2 \\
&\leq \sum_{t\in[T]} \left\{ \left\| P_{\tilde{\phi}_h(\mathcal{D}_h^{N_f,t})}^\perp Q_{\hat{P},h,r^{h,t}}^{\pi^t}(\mathcal{D}_h^{N_f,t}) \right\|^2 + \left\| P_{\tilde{\phi}_h(\mathcal{D}_h^{N_f,t})}^\perp \left( Q_{P^\star,h,r^{h,t}}^{\pi^t}(\mathcal{D}_h^{N_f,t}) - Q_{\hat{P},h,r^{h,t}}^{\pi}(\mathcal{D}_h^{N_f,t}) \right) \right\|^2 \right\} \\
&\overset{(i)}{\leq} \sum_{t\in[T]} \epsilon C_{\min} N_f T + \sqrt{\frac{TN_f\log\frac{2}{\delta}}{2}} + \sum_{t\in[T]} \sigma_1^2(P_{\tilde{\phi}_h(\mathcal{D}_h^{N_f,t})}^\perp)\|Q_{P^\star,h,r^{h,t}}^{\pi^t}(\mathcal{D}_h^{N_f,t})) - Q_{\hat{P},h,r^{h,t}}^{\pi^t}(\mathcal{D}_h^{N_f,t})\|^2 \\
&\overset{(ii)}{\leq} 2\epsilon C_{\min} N_f T + \sqrt{2TN_f\log\frac{2}{\delta}},
\end{aligned}
$$

where $(i)$ follows from $\|Av\|_2 \leq \sigma_1(A)\|v\|_2$ and $(ii)$ follows from that $\sigma_1(P_{\tilde{\phi}_h(\mathcal{D}_h^{N_f,t})}^\perp) \leq 1$ and the process to derive Equation (26).

Table 1: Comparison among provably efficient RL algorithms under low-rank MDPs.

| METHODS | SETTING | SAMPLE COMPLEXITY |
|---|---|---|
| FLAMBE (AGARWAL ET AL., 2020) | LOW-RANK MDP | $\tilde{O}(\frac{H^{22}K^9d^7}{\epsilon^{10}})$ |
| MOFFLE (MODI ET AL., 2021) | LOW-NONNEGATIVE-RANK | $\tilde{O}(\frac{H^5 d_{LV}^3 K^5}{\epsilon^2 \eta})$ |
| HOMER (MISRA ET AL., 2020) | BLOCK MDP | $\tilde{O}(d^8 H^4 K^4(\frac{1}{\epsilon^2} + \frac{1}{\eta^3}))$ |
| RAFFLE (OURS) | LOW-RANK MDP | $\tilde{O}(\frac{H^3 d^2 K(d^2+K)}{\epsilon^2})$ |

[1] We do not include reward-known RL under low-rank MDPs in this table and only focus reward-free RL. The detailed discussion of reward-known RL under low-rank MDPs can be found in Section 7.

We finally use the technique in Du et al. (2021b) to derive the super population guarantee.

$$2\epsilon C_{\min}N_f T + \sqrt{2TN_f \log \frac{2}{\delta}}$$

$$\geq \sum_{t\in[T]} \left\| P^{\perp}_{\tilde{\phi}_h(\mathcal{D}_h^{N_f,t})} \phi_h^{\star}(\mathcal{D}_h^{N_f,t}) w_h^{t\,\star} \right\|_F^2$$

$$= \left\| \left( I - \tilde{\phi}_h(\mathcal{D}_h^{N_f,t}) \left( \tilde{\phi}_h(\mathcal{D}_h^{N_f,t})^\top \tilde{\phi}_h(\mathcal{D}_h^{N_f,t}) \right)^\dagger \tilde{\phi}_h(\mathcal{D}_h^{N_f,t})^\top \right) \phi_h^{\star}(\mathcal{D}_h^{N_f,t}) w_h^{t\,\star} \right\|_F^2$$

$$= \sum_{t\in[T]} (w_h^{t\,\star})^\top \phi_h^{\star}(\mathcal{D}_h^{N_f,t})^\top \left( I - \tilde{\phi}_h(\mathcal{D}_h^{N_f,t})(\tilde{\phi}_h(\mathcal{D}_h^{N_f,t})^\top \tilde{\phi}_h(\mathcal{D}_h^{N_f,t}))^\dagger \tilde{\phi}_h(\mathcal{D}_h^{N_f,t})^\top \right) \phi_h^{\star}(\mathcal{D}_h^{N_f,t}) w_h^{t\,\star}$$

$$= \sum_{t\in[T]} N_f(w_h^{t\,\star})^\top D_{\mathcal{D}_h^{N_f,t}}(\phi_h^{\star}, \tilde{\phi}_h) w_h^{t\,\star}$$

$$\overset{(i)}{\geq} 0.9 \sum_{t\in[T]} N_f(w_h^{t\,\star})^\top D_{q_h}(\phi_h^{\star}, \tilde{\phi}_h) w_h^{t\,\star}$$

$$= 0.9 N_f \sum_{t\in[T]} \left\| D_{q_h}(\phi_h^{\star}, \tilde{\phi}_h)^{1/2} w_h^{t\,\star} \right\|^2$$

$$\geq 0.9 N_f \left\| D_{q_h}(\phi_h^{\star}, \tilde{\phi}_h)^{1/2} W_h^{\star} \right\|_F^2$$

$$\overset{(ii)}{\geq} 0.9 N_f \left\| D_{q_h}(\phi_h^{\star}, \tilde{\phi}_h)^{1/2} \right\|_F^2 \sigma_d^2(W_h^{\star})$$

$$\geq 0.9 N_f \left\| D_{q_h}(\phi_h^{\star}, \tilde{\phi}_h)^{1/2} \right\|_F^2 \frac{C_D T}{d},$$

where $(i)$ follows from lemma B.1 in Du et al. (2021b) and $N_f$ can be large enough, and $(ii)$ follows from that $\|AB\|_F^2 \geq \|A\|_F^2 \sigma_{\min}^2(B)$, where $\sigma_{\min}(B)$ denote the smallest singular value of matrix $B$. Then

$$\left\| D_{q_h}(\phi_h^{\star}, \tilde{\phi}_h)^{1/2} \right\|_F^2 \leq \frac{2\epsilon d C_{\min}}{0.9 C_D} + \frac{d}{0.9 C_D} \sqrt{\frac{2\log \frac{2}{\delta}}{TN_f}}.$$

$\square$

# E  MORE DISCUSSION ON RELATED WORK

In this section, we first summarize the directly comparable work in Appendix E.

## E.1  LOW-RANK MDPS IN EXTENDED RL SETTINGS

Many studies have been developed on various extended low-rank models. Wang et al. (2022); Uehara et al. (2022a) studied partially observable Markov decision process (POMDP) with latent low-rank

structure. Zhan et al. (2022) studied predictive state representations model, and applied their results to POMDP with latent low-rank structure. Cheng et al. (2022); Agarwal et al. (2022) studied benefits of multitask representation learning under low-rank MDPs. Huang et al. (2022) proposed a general safe RL framework and instantiated it to low-rank MDPs. Ren et al. (2022) studied reward-known RL under low-rank MDPs and proposed a spectral method to replace the computation oracle. We note that even given those further developments, our results on lower bound and representation learning are still completely new, and our algorithm design and result on sample complexity are so far still the best-known result for standard reward-free low-rank MDPs.

### E.2 DISCUSSION ON OPTIMISTIC MLE-BASED APPROACH FOR DIFFERENT SETTINGS

There is some concurrent work also using an optimistic MLE-based approach for different settings (POMDP) (Liu et al., 2022; Chen et al., 2022a). We elaborate the key differences between our paper and Liu et al. (2022); Chen et al. (2022a) as follows.

- The two POMDP papers mentioned above study reward-known setting, whereas our focus here is on the reward-free setting.

- Although optimistic MLE-based approach is used in both settings, the design of exploration policy in the two settings are different. Our algorithm identifies which estimated model is used for exploration policy design and then calculate the exploration policy based on bonus terms designed for the value function. However, the POMDP papers construct a confidence set about the true model and solve an optimization problem within this generic model set. Such an oracle may not be easy to realize computationally.

- Due to the hardness of POMDP, MLE approach only guarantees that the estimated distribution of the trajectories is close to the true one. In contrast, in low-rank MDP we study here, we show that the estimation error for the transition probabilities is controlled at each time step.

## F    AUXILIARY LEMMAS

Recall $f_h^{(n)}(s,a) = \|\hat{P}_h^{(n)}(\cdot|s,a) - P_h^\star(\cdot|s,a)\|_{TV}$ represents the estimation error in the $n$-th iteration at step $h$, given state $s$ and action $a$, in terms of the total variation distance. By using Theorem 21 in Agarwal et al. (2020), we are able to guarantee that under all exploration policies, the estimation error can be bounded with high probability.

**Lemma 8.** (MLE guarantee). *Given $\delta \in (0,1)$, we have the following inequality holds for any $n, h \geq 2$ with probability at least $1 - \delta/2$:*

$$\sum_{\tau=0}^{n-1} \mathop{\mathbb{E}}_{\substack{s_{h-1} \sim (P^\star, \pi_\tau) \\ (a_{h-1}, a_h) \sim \mathcal{U}(\mathcal{A}) \\ s_h \sim P^\star(\cdot|s_{h-1}, a_{h-1})}} \left[ f_h^n(s_h, a_h)^2 \right] \leq n\zeta_n, \quad \text{where } \zeta_n := \frac{\log\left(2|\Phi||\Psi|nH/\delta\right)}{n}.$$

*In addition, for $h = 1$,*

$$\sum_{\tau=0}^{n-1} \mathop{\mathbb{E}}_{a_1 \sim \mathcal{U}(\mathcal{A})} \left[ f_1^n(s_1, a_1)^2 \right] \leq n\zeta_n.$$

Divide both sides of the result of lemma Lemma 8 by $n$, and define $\Pi_n = \mathcal{U}(\pi_1, \ldots, \pi_{n-1})$, we have the following corollary which will be intensively used in the analysis.

**Corollary 2.** *Given $\delta \in (0,1)$, the following inequality holds for any $n, h \geq 2$ with probability at least $1 - \delta/2$:*

$$\mathop{\mathbb{E}}_{\substack{s_{h-1} \sim (P^\star, \Pi_n) \\ a_{h-1}, a_h \sim \mathcal{U}(\mathcal{A}) \\ s_h \sim P^\star(\cdot|s_{h-1}, a_{h-1})}} \left[ f_h^n(s_h, a_h)^2 \right] \leq \zeta_n.$$

*In addition, for $h = 1$,*

$$\mathop{\mathbb{E}}_{a_1 \sim \mathcal{U}(\mathcal{A})} \left[ f_1^n(s_1, a_1)^2 \right] \leq \zeta_n.$$

The following lemma (Dann et al., 2017) will be useful to measure the difference between two value functions under two MDPs and reward functions. We define $P_h V_{h+1}(s_h, a_h) = \mathbb{E}_{s \sim P_h(\cdot|s_h,a_h)}[V(s)]$ for shorthand.

**Lemma 9.** (Simulation lemma). *Suppose $P_1$ and $P_2$ are two MDPs and $r_1$, $r_2$ are the corresponding reward functions. Given a policy $\pi$, we have,*

$$V_{h,P_1,r_1}^\pi(s_h) - V_{h,P_2,r_2}^\pi(s_h)$$

$$= \sum_{h'=h}^{H} \mathbb{E}_{\substack{s_{h'} \sim (P_2,\pi) \\ a_{h'} \sim \pi}} \left[ r_1(s_{h'}, a_{h'}) - r_2(s_{h'}, a_{h'}) + (P_{1,h'} - P_{2,h'}) V_{h'+1,P_1,r}^\pi(s_{h'}, a_{h'}) | s_h \right]$$

$$= \sum_{h'=h}^{H} \mathbb{E}_{\substack{s_{h'} \sim (P_1,\pi) \\ a_{h'} \sim \pi}} \left[ r_1(s_{h'}, a_{h'}) - r_2(s_{h'}, a_{h'}) + (P_{1,h'} - P_{2,h'}) V_{h'+1,P_2,r}^\pi(s_{h'}, a_{h'}) | s_h \right].$$

The following lemma is a standard inequality in regret analysis for linear models in reinforcement learning (see Lemma G.2 in Agarwal et al. (2020) and Lemma 10 in Uehara et al. (2021)).

**Lemma 10.** (Elliptical potential lemma). *Consider a sequence of $d \times d$ positive semidefinite matrices $X_1, \ldots, X_N$ with $\mathrm{tr}(X_n) \leq 1$ for all $n \in [N]$. Define $M_0 = \lambda_0 I$ and $M_n = M_{n-1} + X_n$. Then*

$$\sum_{n=1}^{N} \mathrm{tr}\left(X_n M_{n-1}^{-1}\right) \leq 2 \log \det(M_N) - 2 \log \det(M_0) \leq 2d \log\left(1 + \frac{N}{d\lambda_0}\right).$$

If we choose any subset of the set $\{X_n M_{n-1}^{-1}\}_{n=1}^{N}$, we can still get a sublinear summation as follows.

