# OpenReview forum: "Improved Sample Complexity for Reward-free Reinforcement Learning under Low-rank MDPs"
_ICLR.cc/2023/Conference — ICLR 2023 poster_

### Official Review · Reviewer_kHZp · 2022-10-12

**Confidence:** 4
**Correctness:** 4
**Technical Novelty And Significance:** 4
**Empirical Novelty And Significance:** 2
**Recommendation:** 6

**Clarity, Quality, Novelty And Reproducibility:**

The exposition is clear and the results look sound to me.
The novelty of this paper may require more support, as a I pointed out in the weakness part.

**Strength And Weaknesses:**

Pros:
-The paper achieves the SOTA upper and lower bounds for reward-free learning in low rank MDP. This enhances the understanding of low rank MDP of the RL theory community.

Cons:
-I think the authors may want to emphasize more on the difference between different setup in function approximation. For example, citing the Bellman rank paper [1] for the setting of low-rank MDP may not be very accurate. As I understand low-rank MDP is a special case of low Bellman-rank MDP? As a result, comparing results in different setting may involve further complications. The authors may want to elaborate that a little bit.
- The table given is very clear. It would be even better if the sample complexities of algorithms in the normal learning setting can also be compared with the proposed algorithm. This will be helpful as it showcase the difference between the normal setting and reward-free setting.
- It's not clear technique-wise what's the new ingredient that makes the proposed algorithm performs better then the existing ones. The novel termination rule is explained but I believe it is only part of the story. More concretely, it makes me think of the algorithm is a combination of the technique from zero-reward exploration in tabular MDP ([2][3]) and the MLE techniques in FLAMBE. Am I missing anything important here?
- There is some concurrent work also using an optimistic MLE-based approach for different settings (POMDP), such as [4][5]. I know this is not necessary, but it would be a plus if you could also compare it with the algorithmic design and proof technique in these papers.

[1] Contextual Decision Processes with Low Bellman Rank are PAC-Learnable
[2] Task-agnostic Exploration in Reinforcement Learning
[3]  A Sharp Analysis of Model-based Reinforcement Learning with Self-Play.
[4] Optimistic MLE—A Generic Model-based Algorithm for Partially Observable Sequential Decision Making
[5] Partially Observable RL with B-Stability: Unified Structural Condition and Sharp Sample-Efficient Algorithms

**Summary Of The Paper:**

This paper studies the sample complexity of reward-free learning in low-rank MDP and achieves sharp upper and lower bounds. The learned representations can also be reused in a different MDP.

**Summary Of The Review:**

The paper achieves the SOTA upper and lower bounds for reward-free learning in low rank MDP. This enhances the understanding of low rank MDP of the RL theory community. Certain clarification of technical contribution will be super useful to further improve the quality of the paper.

---

> ### Author Response · Authors · 2022-11-18
> **Response to Reviewer kHZp, Part 1**
>
> We thank the reviewer for providing the helpful review! We have addressed the reviewer’s helpful comments and modified the paper accordingly. Please note that in the revised paper, we highlighted our changes by blue-colored texts in both the main body and the appendices of the paper.
>
> Q1: I think the authors may want to emphasize more on the difference between different setup in function approximation. For example, citing the Bellman rank paper [1] for the setting of low-rank MDP may not be very accurate. As I understand low-rank MDP is a special case of low Bellman-rank MDP? As a result, comparing results in different setting may involve further complications. The authors may want to elaborate that a little bit.
>
> A1: Thanks for your valuable suggestions. Yes, low Bellman-rank MDPs can specialize to low-rank MDPs with sample complexity that may have sharper dependence on $d,K$ or $H$, but these algorithms are not computationally efficient as noted in Section 2 in [6]. So they are not directly comparable with our result. We have removed low Bellman-rank MDPs from Table 1 and added a table note to discuss them in Table 1.
>
>
>
> Q2: The table given is very clear. It would be even better if the sample complexities of algorithms in the normal learning setting can also be compared with the proposed algorithm. This will be helpful as it showcase the difference between the normal setting and reward-free setting.
>
> A2: As suggested, we have added the normal setting (Rep-UCB[6] corresponding to the reward-known setting) into our Table 1. In addition, our result on reward-free RL naturally achieves the goal of reward-known RL and improves the sample complexity of [6] by a factor of $ \Theta(H^2)$, which is due to our specially designed truncated value function.
>
>
> Q3: It's not clear technique-wise what's the new ingredient that makes the proposed algorithm performs better then the existing ones. The novel termination rule is explained but I believe it is only part of the story. More concretely, it makes me think of the algorithm is a combination of the technique from zero-reward exploration in tabular MDP ([2][3]) and the MLE techniques in FLAMBE. Am I missing anything important here?
>
> A3: Thanks for the question. Below we elaborate our difference from the studies mentioned above as well as the novel analysis techniques we devise here. All these new developments will clearly demonstrate that our algorithm/analysis goes far beyond the combination of reward-free tabular MDP and FLAMBE.
>
> * **Difference from reward-free tabular MDP:** Although [2] [3] also designed bonus terms with zero-reward to guidance exploration, the bonus terms designed for tabular MDPs are greatly different from the bonus in RAFFLE designed for low-rank MDPs. Intuitively speaking, in the tabular case, the bonus terms $\bar{E}$ can serve as a uniform point-wise error upper bound (uncertainty) for each state-action pair, while in low-rank MDPs, individual bonus term at each step of RAFFLE can not serve as a point-wise error upper bound. Rather, only the truncated value function can serve as a policy-wise error upper bound (uncertainty) for a total trajectory w.r.t each policy. These differences also require different analyses under low-rank MDPs.
>
> * **Difference from MLE in FLAMBE:**
> RAFFLE is distinguished from FLAMBE mainly by two features: a) RAFFLE uses each exploration policy to collect only one sample trajectory in each episode, whereas FLAMBE collects a large number of samples for each episode. Thus, the MLE guarantee of RAFFLE requires significantly fewer samples. b) RAFFLE features a termination step towards the end of exploration phase, which has not been used in FLAMBE.
>
> * **Novel analysis techniques:** *  (1) We specially design truncated value function, which contributes to our sharper result here with a factor of $H^2$ improvement even  compared to the reward-known result [6].
>
> * (2) **Lower bound.**
> We have established a lower bound for reward-free RL under low-rank MDPs. To the best of our knowledge, this is the first lower bound for such a setting. More importantly, we show the dependence of the number of actions is unavoidable, indicating that low-rank MDPs is strictly harder than linear MDPs.
>
> * (3) **Performance guarantee on representation.** We exploit the learned transition kernel from RAFFLE to further learn near-accurate representation. We also theoretically characterize the closeness between the learned and the ground truth representation. This pre-trained representation can be applied in transfer learning to other tasks that share the same representation.

---

> > ### Author Response · Authors · 2022-11-18
> > **Response to Reviewer kHZp, Part 2**
> >
> > Q4: There is some concurrent work also using an optimistic MLE-based approach for different settings (POMDP), such as [4][5]. I know this is not necessary, but it would be a plus if you could also compare it with the algorithmic design and proof technique in these papers.
> >
> > A4: Thanks for pointing out the two studies on POMDP, which are really interesting. We have cited the two papers and included the discussion in the Appendix E in the revision. Below, we elaborate the key differences between our paper and the two POMDP papers as follows.
> >
> > * The two POMDP papers mentioned above study reward-known setting, whereas our focus here is on the reward-free setting.
> >
> > * Although optimistic MLE-based approach is used in both settings, the design of exploration policy in the two settings are different. Our algorithm identifies which estimated model is used for exploration policy design and then calculate the exploration policy based on **bonus terms** designed for the value function . However, the POMDP papers construct a confidence set about the true model and solve an optimization problem within this generic model set. Such an oracle may not be easy to realize computationally.
> >
> >
> >
> > * Due to the hardness of POMDP, MLE approach only guarantees that the estimated distribution of the trajectories is close to the true one. In contrast, in low-rank MDP we study here, we show that the estimation error for the transition probabilities is controlled at each time step.
> >
> > [1] Contextual Decision Processes with Low Bellman Rank are PAC-Learnable.
> >
> > [2] Task-agnostic Exploration in Reinforcement Learning.
> >
> > [3] A Sharp Analysis of Model-based Reinforcement Learning with Self-Play.
> >
> > [4] Optimistic MLE—A Generic Model-based Algorithm for Partially Observable Sequential Decision Making.
> >
> > [5] Partially Observable RL with B-Stability: Unified Structural Condition and Sharp Sample-Efficient Algorithms.
> >
> > [6] Masatoshi Uehara, Xuezhou Zhang, and Wen Sun. Representation learning for online and offline RL in low-rank mdps. In The Tenth International Conference on Learning Representations, ICLR 2022, Virtual Event, April 25-29, 2022. OpenReview.net.
> >
> >
> >
> > Finally, we thank the reviewer again for the helpful comments and suggestions for our work. We are more than happy to address any further questions that you may have during the discussion period.

---

> > > ### Comment · Reviewer_kHZp · 2022-11-18
> > > **thanks for your response!**
> > >
> > > thanks for your response! This answered my questions.

---

### Official Review · Reviewer_rgZQ · 2022-10-23

**Confidence:** 3
**Clarity, Quality, Novelty And Reproducibility:** This paper is well-written and easy t…
**Correctness:** 2
**Technical Novelty And Significance:** 3
**Empirical Novelty And Significance:** Not applicable
**Recommendation:** 3

**Strength And Weaknesses:**

Strength:

The author proposed a novel model-based reward-free (RAFFLE) algorithm and improved the sample complexity in low-rank MDP. In addition, the theoretical lower bound also supports the efficiency of the RAFFLE algorithm.

Weakness:

1. The lower bound on sample complexity (Section 5.2) ignores an essential requirement that $d\ge K$. More specifically, according to figure 1 in Appendix C, the first layer have $L/K$ different states, and the second layer has $L$ states. In this case, the number of different states is at least $K$ since the first layer should have at least one state. Thus, the dimension $d=L\ge K$ and the hard-to-learn instance only hold when $d\ge K$.

2. Based on weakness 1, the claim that reward-free RL under low-rank MDPs is strictly harder than linear MDPs is incorrect. To make this claim, it is necessary to create an instance that shows the lower bound for low-rank MDPs is strictly larger than the upper bound for linear MDPs. However, according to weakness 1, the lower bound only holds for $d\ge K$ (large $d$ regime), and the lower bound does not contradict the $O(d^3H^3/\epsilon^2)$. With a similar argument, we can easily extend the lower bound of linear MDP $O(d^2/\epsilon^2)$ to $O(d|S|/\epsilon^2)$ with the regime $d\ge |S|$ and mentions "linear MDPs is strictly harder than low-rank MDPs since the lower bound to have an additional dependency on $|S|$."

3. The comparison with Uehara et al. (2022b) is incorrect. Uehara et al. (2022b) calculate the sample complexity with the number of steps, while this work measures the sample complexity with the number of trajectories. There always exists a gap of H between trajectories and steps. Thus, the RAFFLE algorithm only improves a factor of $H$ rather than $H^2$.

4. It would be better if the author could provide more intuition on why Lemma 2 holds. The author mentions it is an extension of Lemma 12 in Uehara et al. (2022b) to episodic MDPs. However, in Uehara et al. (2022b), both the triple at stage $h-1$ and state $h$ are used to estimate the model (MLE). It would be better if the author could provide more intuition as to why only using the data at stage $h$ can maintain a similar performance or the uniform choose action $a_{h-1}$ is unnecessary.

5. It is strange why the RAFFLE can always find the correct representation $\phi^*$ in Theorem 4. More specifically, even for a known transition kernel $P$, there may still exist multi-representation for $P$ in the lower-rank MDP. For instance, if we switch the first and second dimensions of $\phi_1,\mu_1$ and obtain $\phi_2,\mu_2$, then two representations will lead to the same transition kernel. Under this situation, it seems impossible to learn the actually representation $\phi^*$ from $\phi_1$ and $\phi_2$.

**Summary Of The Paper:**

This work focused on reward-free RL with low-rank MDP. The author proposed a new model-based reward-free (RAFFLE) algorithm and provided a theoretical guarantee with polynomial sample complexity. In addition, the author offers a lower sample complexity bound for any algorithm within the low-rank MDP. Finally, the author shows it is possible further to learn a provably near-accurate representation of the transition kernel.



**Summary Of The Review:**

This work proposes a novel algorithm (RAFFLE) for learning reward-free low-rank MDPs and improving the sample complexity. However, it seems that the lower bound ignores an essential requirement, and the author makes a wrong claim, "reward-free RL under low-rank MDPs is strictly harder than linear MDPs is incorrect." as a contribution. In addition, I am concerned about why it is possible to distinguish two representations in Theorem 4, when they correspond to the same transition kernel $P$.

---

> ### Author Response · Authors · 2022-11-18
> **Response to Reviewer rgZQ, Part 1**
>
> We thank the reviewer for providing the helpful review! We have addressed the reviewer’s comments and modified the paper accordingly. Please note that in the revised paper, we highlighted our changes by blue-colored texts in both the main body and the appendices of the paper.
>
> Q1: The lower bound on sample complexity (Section 5.2) ignores an essential requirement that $d \geq K$. More specifically, according to figure 1 in Appendix C, the first layer have $L/K$ different states, and the second layer has $L$ states. In this case, the number of different states is at least $K$ since the first layer should have at least one state. Thus, the dimension $d=L\geq K$ and the hard-to-learn instance only hold when $d \geq K$.
>
> A1: Many thanks for this really helpful comments! We agree that the hard instance given in the initial submission holds only when $d \geq K$. Excitingly, in the revision, we were able to develop a new lower bound, which holds for any $d$ and $K$, without requiring $d \geq K$. Our proof of the new lower bound mainly feature the following two novel ingredients in the construction of hard MDP instances:
>
> * a) We divide the actions into two types. The first type of actions is mainly used to form a large state space through a tree structure. The second type of actions is mainly used to distinguish different MDPs. Such a construction allows us to separately treat the state space and the action space, so that both state space and action space can be arbitrarily large.
>
> * b) We explicitly define the feature vectors for all state-action pairs, and more importantly, the dimension is less than or equal to the number of states. Thus, there is no constraint on the relationship between the dimension $d$ and the number of actions $K$. We then show that the lower bound is $\Omega(\frac{HdK}{\epsilon^2})$, which holds without requiring our previous assumption of $d \geq K$. This establishes a solid statement that the low-rank MDPs is harder than linear MDPs in terms of the sample complexity dependence on the number of actions $K$.
>
> We have updated our proof in Appendix C to include the new proof.
>
>
>
> Q2: Based on weakness 1, the claim that reward-free RL under low-rank MDPs is strictly harder than linear MDPs is incorrect. To make this claim, it is necessary to create an instance that shows the lower bound for low-rank MDPs is strictly larger than the upper bound for linear MDPs. However, according to weakness 1, the lower bound only holds for $d \geq K$ (large  $d$ regime), and the lower bound does not contradict the $O(d^3H^3/\epsilon^2)$. With a similar argument, we can easily extend the lower bound of linear MDP $O(d^2/\epsilon^2)$  to $O(d|S|/\epsilon^2)$ with the regime $d \geq |S|$ and mentions "linear MDPs is strictly harder than low-rank MDPs since the lower bound to have an additional dependency on $|S|$."
>
> A2: Please refer to our response A1.
>
>
> Q3: The comparison with Uehara et al. (2022b) is incorrect. Uehara et al. (2022b) calculate the sample complexity with the number of steps, while this work measures the sample complexity with the number of trajectories. There always exists a gap of $H$ between trajectories and steps. Thus, the RAFFLE algorithm only improves a factor of $H$ rather than $H^2$.
>
> A3: We would like to clarify that the sample complexity of Uehara et al. (2022b) ([1]) in fact is in the number of trajectories. As described in Sec 4.1 in Uehara et al. (2022b), to collect each sample $(s,a,s^\prime,a^\prime,\tilde{s})$ in discounted infinite MDP, the agent requires a roll-in, i.e, a trajectory to get into $s$ using certain exploration policy to obtain this sample. In addition, each roll-in terminates in $O(1/1-\gamma)$ steps with high probability, where $\gamma$ is the discounted factor. Therefore, RAFFLE indeed improves a factor of $H^2$ over Uehara et al. (2022b).

---

> > ### Author Response · Authors · 2022-11-18
> > **Response to Reviewer rgZQ, Part 2**
> >
> > Q4: It would be better if the author could provide more intuition on why Lemma 2 holds. The author mentions it is an extension of Lemma 12 in Uehara et al. (2022b) to episodic MDPs. However, in Uehara et al. (2022b), both the triple at stage $h-1$ and state $h$ are used to estimate the model (MLE). It would be better if the author could provide more intuition as to why only using the data at stage $h$ can maintain a similar performance or the uniform choose action $a_{h-1}$ is unnecessary.
> >
> > A4: Thanks for the comments. The triples at stage $h-1$ in iteration $h$ are generated by running exploration policy up to step $h-1$ and taking an uniform action. By using importance sampling in step $h-2$, the information of triples at stage $h-1$ in iteration $h$ can be covered by the triples that are generated by running exploration policy up to step $h-2$ and taking two uniform actions consecutively, which are exactly the triples at stage $h-1$ in iteration $h-1$ and are also used in MLE. In other words, the information of triples at stage $h-1$ in iteration $h$ can be covered by the information of triples at stage $h-1$ in iteration $h-1$. Thus only using the data at stage $h$ can maintain  similar performance.
> >
> >
> > Q5: It is strange why the RAFFLE can always find the correct representation $\phi^\star$ in Theorem 4. More specifically, even for a known transition kernel $P$, there may still exist multi-representation for $P$ in the lower-rank MDP. For instance, if we switch the first and second dimensions of $\phi_1,\mu_1$ and obtain $\phi_2,\mu_2$, then two representations will lead to the same transition kernel. Under this situation, it seems impossible to learn the actually representation $\phi^\star$ from $\phi_1$ and $\phi_2$.
> >
> > A5: Thanks for the comments. We would like to clarify that the divergence guarantee in Theorem 4 doesn't imply that the agent learns the correct $\phi^\star$ point-wisely. In fact, the divergence guarantee implies that the agent learns a near-accurate **linear space spanned by the representation**.
> > More specifically, the divergence is related to certain data distribution $q$. When we consider the realization of representation $\phi=\{\phi_1,\phi_2,\ldots,\phi_d\}$ on certain dataset $\mathcal{D}=\sum_{i=1}^n(s^i,a^i), \mathcal{D} \sim q$, we can view the realization of each dimension of $\phi$, $(\phi_j(s^1,a^1),\ldots,\phi_j(s^n,a^n))^\top$, $j \in \{1, \ldots,d\}$ as a vector and all of these dimensions of $\phi$, denoted as $\{\phi_1(\mathcal{D}),\ldots,\phi_d(\mathcal{D})\}$, span a linear space.  Then our Theorem 4 implies that the spaces spanned by $\phi^\star$ and $\tilde{\phi}$ w.r.t certain dataset $\mathcal{D}$ are sufficiently close. As a result, although the transition kernel $P$ may have different decompositions, for instance, $\phi_1,\mu_1$ and $\phi_2,\mu_2$, the spaces spanned by $\phi_1$ and $\phi_2$ are the same and the divergence between $\phi_1$ and $\phi_2$ equals to zero.
> >
> > [1] Masatoshi Uehara, Xuezhou Zhang, and Wen Sun. Representation learning for online and offline RL in low-rank mdps. In The Tenth International Conference on Learning Representations, ICLR 2022, Virtual Event, April 25-29, 2022. OpenReview.net.
> >
> >
> > Finally, we thank the reviewer again for the helpful comments and suggestions for our work. If our response resolves your concerns to a satisfactory level, we kindly ask the reviewer to consider raising the score of your evaluation. Certainly, we are more than happy to address any further questions that you may have during the discussion period.

---

> ### Author Response · Authors · 2022-12-01
> **Your feedback is important to us**
>
> Dear Reviewer rgZQ,
>
> This is a friendly reminder that we have submitted our response to your review comments and uploaded a revision of the paper two weeks ago, and we will appreciate very much if you could give us any feedback. In particular, regarding your main concern of the lower bound, our revision of the paper has developed a new proof of the lower bound with full technical steps, which supported our previous statement. Our response further explained in detail about our ideas of the proof. Our response also provided answers to several other questions you asked in your review. If our response resolves your concerns, we kindly ask you to consider raising the rating of our work. We are also more than happy to answer your further questions. Thank you very much for your time and efforts!

---

### Official Review · Reviewer_2hcZ · 2022-10-24

**Confidence:** 4
**Clarity, Quality, Novelty And Reproducibility:** The paper is well-written overall.
**Correctness:** 4
**Technical Novelty And Significance:** 2
**Empirical Novelty And Significance:** 2
**Recommendation:** 5

**Strength And Weaknesses:**

Strength: propose a new model-based reward-free RL algorithm under low-rank MDPs. The algorithm can both find an ε-optimal policy and achieve an ε-accurate system identification via reward-free exploration, with a sample complexity of $\tilde{O}(H^3d^2K(d^2+K))$. A lower bound on the sample complexity that holds for any reward-free algorithm under low-rank MDPs.

Weakness:

The main weakness of this paper is the algorithm seems close related to [1]. Note reward-free multi-task is a more general setting than the current study and the exploration of RAFFLE seems to be very similar to REFUEL in [1] (with $T=1$). In addition, the main contribution of this paper seems to have overlap with [1]. For instance, (6) of Theorem 4.2. is a system identification result and (7) is the reward-free learning result. At this moment, I suspect the result of the current paper might be a corollary of [1]. It is likely I might be wrong, so please correct me if I misunderstood something.

[1] Provable benefit of multitask representation learning in reinforcement learning, NeurIPS 2022




**Summary Of The Paper:**

This paper proposes a new model-based algorithm, coined RAFFLE, and show that it can both find an $\epsilon$-optimal policy and achieve an $\epsilon$-accurate system identification via reward-free exploration, with
a sample complexity of $\tilde{O}(H^3d^2K(d^2+K))$, where $d,H$ and $K$ respectively denote dimension, episode horizon, and action space cardinality. This significantly improves the sample complexity in Agarwal et al. (2020) for the same learning goals.

We further provide a sample complexity lower bound that holds for any reward-free algorithm under low-rank MDPs, which matches our upper bound in the dependence on $\epsilon$, as well as on $K$ in the large $d$ regime.

**Summary Of The Review:**

Please answer my major concern.

---

> ### Author Response · Authors · 2022-11-18
> **Response to Reviewer 2hcZ**
>
> We thank the reviewer for providing the helpful review! We have addressed the reviewer’s comments and modified the paper accordingly. Please note that in the revised paper, we highlighted our changes by blue-colored texts in both the main body and the appendices of the paper.
>
> Q1: The main weakness of this paper is the algorithm seems close related to [1]. Note reward-free multi-task is a more general setting than the current study and the exploration of RAFFLE seems to be very similar to REFUEL in [1] (with $T=1$). In addition, the main contribution of this paper seems to have overlap with [1]. For instance, (6) of Theorem 4.2. is a system identification result and (7) is the reward-free learning result. At this moment, I suspect the result of the current paper might be a corollary of [1]. It is likely I might be wrong, so please correct me if I misunderstood something.
>
> [1] Provable benefit of multitask representation learning in reinforcement learning, NeurIPS 2022.
>
> A1: Thanks for the comments. Although reward-free multi-task RL in [1] is a more general setting than our setting here, the result on sample complexity that we have here is sharper than that instantiated from [1]. Besides, we have provided more results in this work, such as a lower bound and the performance guarantee on learned representation, which are not covered in [1]. We next elaborate further about these differences.
>
>
> * (1) **Sample complexity.** Our algorithm RAFFLE achieves the system identification and reward-free learning with sample complexity of $\tilde{O}(\frac{H^3d^2K(d^2+K)}{\epsilon^2})$. As a comparison, instantiating the result in [1] to $T=1$ yields the sample complexity of $\tilde{O}(\frac{(H^2+d)(d^2+K)H^3dK}{\epsilon^2})$. Clearly, our result here is sharper with a factor of $H^2$ improvement over [1]. This is due to our specially designed truncated value function, which may not work in the multitask scenario in [1].
>
> * (2) **Performance guarantee on representation.** We exploit the learned transition kernel from RAFFLE to further learn near-accurate representation. We also theoretically characterize the closeness between the learned and the ground truth representations. Such results are not presented in [1]. Instead, [1] makes additional assumptions 4 and 5 to directly build connection between source tasks and target tasks and only shows the closeness of the estimated and true target transitions (rather than representation) as in their Lemma 1. Note that with our result on the guaranteed accurate representation here, it is possible to relax the assumptions for transfer learning in [1].
>
>
> * (3) **Lower bound.**
> We have established a lower bound for reward-free RL under low-rank MDPs. To the best of our knowledge, this is the first lower bound for such a setting. More importantly, we show that the dependence of the number of actions is unavoidable, indicating that low-rank MDPs is strictly harder than linear MDPs. Clearly, [1] does not have such a lower bound result.
>
> To speak a little further about the lower bound, in fact, one of our biggest updates in the revision is that we have
> developed a new improved lower bound, which does not require our previous implicit assumption of $d \geq K$. Our proof of the new lower bound mainly feature the following two novel ingredients in the construction of hard MDP instances:
>
> * a) We divide the actions into two types. The first type of actions is mainly used to form a large state space through a tree structure. The second type of actions is mainly used to distinguish different MDPs. Such a construction allows us to separately treat the state space and the action space, so that both state space and action space can be arbitrarily large.
>
> * b) We explicitly define the feature vectors for all state-action pairs, and more importantly, the dimension is less than or equal to the number of states. Thus, there is no constraint on the relationship between the dimension $d$ and the number of actions $K$. We then show that the lower bound is $\mathcal O(\frac{HdK}{\epsilon^2})$, which holds without requiring our previous assumption of $d \geq K$. This establishes a solid statement that the low-rank MDPs is harder than linear MDPs in terms of the sample complexity dependence on the number of actions $K$.
>
>
> Finally, we thank the reviewer again for the helpful comments and suggestions for our work. If our response resolves your concerns to a satisfactory level, we kindly ask the reviewer to consider raising the score of your evaluation. Certainly, we are more than happy to address any further questions that you may have during the discussion period.

---

> ### Author Response · Authors · 2022-12-01
> **Your feedback is important to us**
>
> Dear Reviewer 2hcZ,
>
> This is a friendly reminder that we have submitted our response to your review comments and uploaded a revision of the paper two weeks ago, and we will appreciate very much if you could give us any feedback. In particular, our response explained in detail our new contributions compared to [1] ("Provable benefit of multitask representation learning in reinforcement learning", NeurIPS 2022). If our response resolves your concern, we kindly ask you to consider raising the rating of our work. We are also more than happy to answer your further questions. Thank you very much for your time and efforts!

---

### Official Review · Reviewer_exQF · 2022-10-24

**Confidence:** 4
**Clarity, Quality, Novelty And Reproducibility:** The paper is overall clear with limit…
**Correctness:** 3
**Technical Novelty And Significance:** 2
**Empirical Novelty And Significance:** Not applicable
**Recommendation:** 5

**Strength And Weaknesses:**

### Strength:
* The idea is simple and easy to follow.
* The interpretation of the lower bound is interesting.

### Weakness:
* The proof of the lower bound lacks some rigor.
* Section 5.3 is not well-motivated under the context of RL. In other words, I don’t quite understand how this divergence will help to analyze the performance of reinforcement learning.


**Summary Of The Paper:**

This paper combines the REP-UCB and RF-UCRL to obtain an improved sample complexity for reward-free reinforcement learning under the structural assumption on Low-rank MDPs.


**Summary Of The Review:**

For me the combination of REP-UCB and RF-UCRL is not so striking. The most interesting part for me is the lower bound, where the authors argue that we cannot avoid the dependency on the number of actions if we need to learn the representation. However, as the proof lacks some rigor, I'm not sure if the proof is correct. For me, intuitively the lower bound is correct, as for the instance the authors consider, if we know the exact feature, we already know which action is different. I believe the authors need to make the proof of the lower bound much more formal. The analysis in Section 5.3 should be much more well-motivated.

---

> ### Author Response · Authors · 2022-11-18
> **Response to Reviewer exQF, Part 1**
>
> We thank the reviewer for providing the helpful review! We have addressed the reviewer’s helpful comments and modified the paper accordingly. Please note that in the revised paper, we highlighted our changes by blue-colored texts in both the main body and the appendices of the paper.
>
> Q1: The proof of the lower bound lacks some rigor.
>
> A1: Thanks for the suggestions. The previous version of the proof omitted some straightforward steps
> for a succinct presentation.
> In fact, in the revision, one of our biggest updates is a new improved lower bound, which does not require our previous implicit assumption of $d \geq K$. For this new lower bound, we have included the detailed steps in the analysis in Appendix C. We hope the reviewer will find it satisfactory.
>
>
> Our proof of the new lower bound mainly lies in the construction of hard MDP instances, which feature the following two novel ingredients.
>
> * a) We divide the actions into two types. The first type of actions is mainly used to form a large state space through a tree structure. The second type of actions is mainly used to distinguish different MDPs. Such a construction allows us to separately treat the state space and the action space, so that both state space and action space can be arbitrarily large.
>
>
> * b) We explicitly define the feature vectors for all state-action pairs, and more importantly, ensure that the dimension is less than or equal to the number of states. Thus, there is no constraint on the relationship between the dimension $d$ and the number of actions $K$. We then show that the lower bound is $\mathcal O(\frac{HdK}{\epsilon^2})$, which holds without requiring our previous assumption of $d \geq K$. This establishes a solid statement that the low-rank MDPs is harder than linear MDPs in terms of the sample complexity dependence on the number of actions $K$.
>
>
> Q2: Section 5.3 is not well-motivated under the context of RL. In other words, I don’t quite understand how this divergence will help to analyze the performance of reinforcement learning.
>
> A2: Many thanks for the great question. (1) Proving a guarantee on the representation presents theoretical significance. (2) Our study of Section 5.3 is well motivated by transfer learning, in which the learned representation is applied later on to other tasks that share the same representation. For instance, [1] requires to attain such a divergence guarantee in their analyses when applying the pre-trained representation to new tasks.
> More specifically, consider a transfer learning scenario where the agent is first assigned with a source task to pre-train representation and then is assigned with a target task that shares the common representation $\phi^\star$ with the source task. It is then important that the performance guarantee on the closeness between $\hat{\phi}$ and $\phi^\star$ is characterized, in order to be able to analyze the performance of the target task.
> It is anticipated that when the divergence is sufficiently small, the pre-trained representation can help to improve transfer learning efficiency.
> We remark here that our Section 5.3 is still different with [1] because [1] assumes a generative model to generate data, while our data generation is based on $\hat{P}$ from RAFFLE.

---

> > ### Author Response · Authors · 2022-11-18
> > **Response to Reviewer exQF, Part 2**
> >
> > Q3: RAFFLE is a the combination of REP-UCB and RF-UCRL and is not so striking.
> >
> > A3: We respectfully disagree with the above comments, as we elaborate below.
> >
> > **Compared with RF-UCRL ([2])**: In the tabular case, the bonus terms $\bar{E}$ designed for tabular MDPs in [2] are greatly different from the bonus $\hat{b}$ designed for low-rank MDPs in RAFFLE. Intuitively, in the tabular case, the bonus term $\bar{E}$ can serve as a uniform point-wise error upper bound (uncertainty) for each state-action pair. However, in low-rank MDPs, individual bonus term at each step of RAFFLE cannot serve as a point-wise error upper bound. Only the truncated value function can serve as a policy-wise error upper bound (uncertainty) for a total trajectory w.r.t each policy. These differences also require greatly different analyses under low-rank MDPs.
> >
> > **Algorithmic comparison with REP-UCB([3]):** Algorithmically, RAFFLE features three new designs different from REP-UCB:
> >
> > * a) Different exploration principle: as a **reward-free** algorithm, RAFFLE uses bonus term $\hat{b}^{(n)}$ and truncated value function to explore state-action where model estimation has large uncertainty. REP-UCB focuses on **reward-known** setting, and uses $\hat{b}^{(n)}$ for optimistic action selection to maximize value function for the given reward.
> >
> > * b) Unique termination condition, which ensures the accuracy of the model estimate $\hat{P}^{(n)}$, and further guarantees near-optimal policy under *any reward* function. In contrast, REP-UCB doesn't have this termination condition, hence does not have guarantee on model identification. It only returns a near-optimal policy for the *given reward*.
> >
> > * c) We specially design truncated value function, which contributes to our sharper result here with a factor of $H^2$ improvement over REP-UCB.
> >
> > **Technical comparison with REP-UCB([3]):** The three new designs necessitate *new analysis* for proving Theorems 1 and 2.
> >
> > * a) Step 2 (Proposition 2) in Appendix A for proving Theorem 1 develops new bounds for reward-free setting under low-rank MDPs (due to 1st new design).
> >
> > * b) Step 3 (Proposition 3) in Appendix A for proving Theorem 1 and proof of Theorem 2 develop new analysis to capture benefit of the termination condition to RAFFLE (2nd new design).
> >
> > * c) Step 1, 2 and 3 (Proposition 1, 2 and 3) in Appendix A all develop new analyses to handle our specially designed truncated value function, whereas [3] does not contain such analyses.
> >
> > [1] Rui Lu, Gao Huang, and Simon S Du. On the power of multitask representation learning in linear mdp. arXiv preprint arXiv:2106.08053, 2021.
> >
> > [2] Emilie Kaufmann, Pierre Ménard, Omar Darwiche Domingues, Anders Jonsson, Edouard Leurent, Michal Valko. Adaptive Reward-Free Exploration. ALT 2021: 865-891.
> >
> > [3] Masatoshi Uehara, Xuezhou Zhang, and Wen Sun. Representation learning for online and offline RL in low-rank mdps. In The Tenth International Conference on Learning Representations, ICLR 2022, Virtual Event, April 25-29, 2022. OpenReview.net.
> >
> > Finally, we thank the reviewer again for the helpful comments and suggestions for our work. If our response resolves your concerns to a satisfactory level, we kindly ask the reviewer to consider raising the score of your evaluation. Certainly, we are more than happy to address any further questions that you may have during the discussion period.

---

> ### Author Response · Authors · 2022-12-01
> **Your feedback is important to us**
>
> Dear Reviewer exQF,
>
> This is a friendly reminder that we have submitted our response to your review comments and uploaded a revision of the paper two weeks ago, and we will appreciate very much if you could give us any feedback. In particular, regarding your main concern of the lower bound, our revision of the paper has developed a new proof of the lower bound with full technical steps, and our response explained in detail about our ideas in the proof. Our response also clarified our novelty compared to the previous algorithms that you mentioned. If our response resolves your concerns, we kindly ask you to consider raising the rating of our work. We are also more than happy to answer your further questions. Thank you very much for your time and efforts!

---

### Decision · Program_Chairs · 2023-01-20

**Decision:**

Accept: poster

**Justification For Why Not Higher Score:**


Although the sample complexity improvement is significant, the technique is largely borrowed from RepUCB, which is not novel.

**Justification For Why Not Lower Score:**


The lower bound is interesting that demonstrates the difference between low-rank MDPs and linear MDPs.

**Metareview: Summary, Strengths And Weaknesses:**


In this paper, the authors improved the sample complexity for reward-free reinforcement learning under the low-rank MDPs setting with a model-based method. They further investigated the lower bound of the sample complexity, demonstrating the essential difficulty w.r.t. the linear MDP.

This indeed a borderline paper. Although the sample complexity for reward-free RL under low-rank MDP has been improved significantly, the imporovement is as expected with the techniques in RepUCB and the algorithm is an extension of the RF-UCRL, which are not novel.

The major novelty of the paper is the new lower bound, which reveals the essential difficulty comparing to the linear MDPs. This part makes the paper worth to be published if it is correct. However, as reviwer rgZQ pointed out, the lower bound in the first version of the submission requiring d\ge K, which will make the lower bound trivial. The authors eventually fixed this in the second version, but the correctness of this updated proof has not been checked by reviewer rgZQ. Based on the discussion with reviewer exQF, the reviewer believed the claim is correct, and would like to raise the score to 6 (while the system is already locked).

Based on these discussion, I personally think it is okay to accept the paper if there is space, but I am also fine if the decision is bumped down.

**Note From Pc:**

if the above contains the word "oral" or "spotlight" please see: "oral" presentation means -> notable-top-5% and "spotlight" means -> notable-top-25%. As stated in our emails, we are disassociating presentation type from AC recommendations

**Summary Of Ac-Reviewer Meeting:**


Based on personal communication with Reviewer exQF, the lower bound in the update version makes sense and interesting. The reviewer would like to increase the score to 6, but the openreview system is already locked.